# Chemo-mechanical forces modulate the topology dynamics of mesoscale DNA assemblies

Deepak Karna [1], Eriko Mano[2], Jiahao Ji [1], Ibuki Kawamata [3] ✉, Yuki Suzuki [2,4] ✉ & Hanbin Mao [1] ✉

The intrinsic complexity of many mesoscale (10–100 nm) cellular machineries makes it challenging to elucidate their topological arrangement and transition dynamics. Here, we exploit DNA origami nanospring as a model system to demonstrate that tens of piconewton linear force can modulate higher-order conformation dynamics of mesoscale molecular assemblies. By switching between two chemical structures (i.e., duplex and tetraplex DNA) in the junctions of adjacent origami modules, the corresponding stretching or compressing chemo-mechanical stress reversibly flips the backbone orientations of the DNA nanosprings. Both coarse-grained molecular dynamics simulations and atomic force microscopy measurements reveal that such a backbone conformational switch does not alter the right-handed chirality of the nanospring helix. This result suggests that mesoscale helical handedness may be governed by the torque, rather than the achiral orientation, of nanospring backbones. It offers a topology-based caging/uncaging concept to present chemicals in response to environmental cues in solution.

Mesoscale assemblies have the size on the order of 10 nm to 500 nm[1]. In the biological context, the mesoscale dimensions hold an important locus as most viral particles and many important cellular machineries are mesoscale sized biomolecular assemblies. However, due to the molecular complexity of mesoscale assemblies, the lack of model systems, and limited characterization techniques, principles governing biological mesoscale structures are not fully understood[2]. Given that mesoscale bioassemblies assume critical biological functions, it becomes urgent to elucidate structural organization principles in mesoscale structures. Due to its programmable nature, we anticipate DNA origami nanoassemblies[3,4] serve a readily accessible model to investigate principles of topological arrangements in mesoscale structures. A typical DNA origami nanoassembly employs conventional Watson-Crick base pairing in which several DNA duplexes are bundled together to form a 2D or 3D nanostructures[4–7]. The method is assisted by computer aided designs to simulate hybridization of several tens to hundreds of single-stranded small DNA fragments, called staples, onto a long single-stranded scaffold template DNA[3,8]. Such a one-pot annealing reaction readily synthesizes higher-order nano and mesoscale structures. With the precise and specific base pairing in DNA duplexes and supramolecular nature of DNA origami self-assembly, the method provides ample space to introduce different functional groups. Among topological organizations at different length scales, it becomes especially relevant to divulge the structural-property relationship as well as modulation factors behind the long-range, higher-order arrangement of subunits in a mesoscale assembly. These higher-order spatial arrangements include backbone topology of the mesoscale structure, which determines overall conformation such as spheres and springs[9–11], as well as specific interactions between local structural components and solvent molecules. Numerous applications

[1]Department of Chemistry and Biochemistry, Kent State University, Kent, OH 44242, USA. [2]Frontier Research Institute for Interdisciplinary Sciences, Tohoku University, 6-3 Aramaki-aza Aoba, Aoba-ku, Sendai 980-8578, Japan. [3]Department of Robotics, Graduate School of Engineering, Tohoku University, 6-6-01 Aramaki-aza Aoba, Aoba-ku, Sendai 980-8579, Japan. [4]Department of Chemistry for Materials, Graduate School of Engineering, Mie University, 1577 Kurimamachiya-Cho, Tsu 514-8507, Japan. ✉e-mail: ibuki.kawamata@tohoku.ac.jp; ysuzuki@chem.mie-u.ac.jp; hmao@kent.edu

arise after mesoscale conformations can be harnessed. For example, when a backbone topology is responsive to external cues, a new, conformation-based uncaging mechanism can be configured if the response brings enclosed chemicals to external surfaces.

The topology in mesoscale backbone may also set the higher-order chirality of mesoscale biological structures. Chirality is a universal phenomenon in both biotic and abiotic worlds. In biotic system, the chirality determines the activity of enzymes toward the substrate with a matching chiral sense. This reinforcement in chirality selection is one of the reasons causing the homochirality[12] on earth. At the atomic level, chirality is originated from the arrangement of four different functional groups in a tetrahedral space surrounding a central atom. Similar arrangement of microscopic or macroscopic objects leads to opposite chiralities represented by non-overlapping mirror symmetries. For biomacromolecules, secondary structures such as left-handed or right-handed helices form nanoscopic chiralities. Examples include DNA double/triple helices and peptide coiled coils. At this level, it has been shown that chirality transmission exists between different nanoscopic helicities with diameters smaller than 10 nm[13], which may serve to govern the interaction between two protein molecules or between nucleic acids and proteins.

For mesoscale protein structures, higher-order helices with diameters more than 10 nm exist in filaments made of polymerized myosin molecules for example[14]. For mesoscale DNA origami structures, the length of duplex DNA can be micrometer or longer. However, the diameter of duplex DNA with left-handed or right-handed helicity is nanoscopic (≈2 nm diameter). Bundles of many dsDNA strands have been demonstrated in the nano- or meso-scale DNA origami self-assemblies with different helicity handedness[15–17]. Intermolecular force (IMF)[18] can induce conformational variation in different parts of a protein, leading to allostery in a nanometer scale[19,20]. However, in mesoscale helices, IMF may not be strong enough to sustain the preferential long-range molecular arrangement across hundreds of nanometers space to produce different helical senses (i.e., left-handed or right-handed twists) in the mesoscopic chirality. Compared to the short range IMF interactions (which scales to few nanometers), mechanical interaction has shown long-range properties[13]. It has been shown that torques in left- or right-handed biomolecular helices (e.g., DNA double helix and peptide coil-coils) can propagate along a distance up to 4.5 nm[13] between a section of DNA double helix and one set of peptide coiled coils. Given that peptide coiled coils have much weaker twisting density than duplex DNA, even longer chirality transmission distance may be found in the coupling between two DNA double helices, which are prevalent in mesoscale DNA origami assemblies. Therefore, it is conceivable that long-range mechanical interaction may play a predominate role in the organization of mesoscale structures.

In this work, we prepared DNA origami nanosprings that contain 37 modules with 37 actuatable junctions, each of which allows the transmission of helical chirality of double-stranded DNA. By formation of duplex or tetraplex DNA structures in each junction between neighboring origami modules, the mechanical bending direction of the DNA nanospring backbone is reversibly switched under tens of pN force. This results in two nanosprings with their backbone orientations flipped while maintaining the same right-handed helicity with 25.8 to 43.9 nm in helical diameters. Therefore, the linear chemo-mechanical force is not sufficient to change the chirality of nanospring helix, which is likely determined by the rotational torque inherent in right-handed DNA double helices[21] constituting the nanospring backbones. Using optical tweezers, we have also found that the spring constants are larger in nanosprings with smaller diameters and shorter spring length, probably because of more compact stacking of nanospring coils. Our work helps to explain the chiral origin of mesoscale helices and provides an example of linear chemo-mechanical modulations on the achiral topology dynamics of mesoscale DNA assemblies.

## Results and discussion
### Preparation and 2D characterization of DNA nanosprings
The dual-switching nanospring is based on the design of our previous nanospring[22,23], which was folded from a circular ssDNA template (p8064) by DNA origami method. The nanospring contained 37 repeats of a transformable module unit comprising a stem, 2 piers, and 2 bridge strands in each junction (Fig. 1a, b and Supplementary Fig. 1, see Supplementary Fig. 2 for detailed origami sequences). The bridge strand contained a human telomeric G-rich DNA repeat sequence (5′-GGGTTAGGGTTAGGGTTAGGG-3′) flanked with staple sequences that were folded into each pier. Upon the G-quadruplex formation induced by K+, the bridge strands contracts, thereby causing the bending of the module (Fig. 1b). Via cumulative effect of this bending, the entire shape is transformed from the relaxed shape into a coiled, spring-like shape (Fig. 1a).

To achieve the actuation by signals other than K+, we designed an anti-GQ strand carrying a toehold (underlined), 5′-CCCTAACCC-TAACCCTAACCCAGAGAACT-3′ (anti-GQ-toe), to hybridize with the GQ-forming bridge by forming a 21 bp duplex DNA. To ensure complete hybridization, we used 1 µM anti-GQ strand, which is about 1000 times higher concentration (see Supplementary Fig. 3 for optimized hybridization ratio) than the effective concentration of single molecules tethered between two trapped particles[24]. The stiff duplex DNA (50 nm persistence length[25]) pushes the piers to bend to the direction opposite to that induced by the GQ-formation (Fig. 1b). The anti-GQ-toe can be displaced via the toehold-mediated strand displacement with its fully complementary releaser strand, 5′-AGTTCTCTGGGTTAGGGTTAGGGTTAGGG-3′, allowing reversible transformation by specific DNA strands. Overall, the nanodevice possesses a dual-responsivity against K+ and DNA fuels and transforms the mesoscale assembly into different shapes depending on external signals.

The reversible transformation by each signal was confirmed by atomic force microscopy (AFM) imaging on a 2D surface, which revealed clear morphological differences in respective conditions (Fig. 1c, Supplementary Figs. 3 and 4). Spirally coiled structures of nanosprings were observed in the presence of K+ or after incubation with anti-GQ strand owing to the cumulative effect of bending modules, while the origami without K+ or anti-strands showed a relaxed linear structure. Compared with the K+-induced nanospring (GQ-NS), the anti-GQ-strand-incorporated nanospring (anti-GQ-NS) took a more coiled and compact structure as reflected in the statistical analyses of the AFM images. Measured values of the radius of the curvature and number of turns for anti-GQ-NS were 25.8 ± 2.8 nm (mean ± SD) and 3.5 ± 0.4, respectively, whereas those for GQ-NS were 43.9 ± 10.3 nm and 1.7 ± 0.4 (Fig. 1d–g, Supplementary Figs. 5 and 6).

### 3D spatio-mechanical properties of DNA nanosprings
To investigate structures and properties of as-synthesized dual switching nanosprings in 3D space, we used optical tweezers to stretch and relax these nanosprings in a 5-channel microfluidic chamber (Fig. 2a). First, we tethered each nanospring between two dsDNA handles, which are anchored to two optically trapped polystyrene beads via biotin-streptavidin and digoxigenin antibody-digoxigenin interactions, respectively.

We then performed force-ramping experiments in optical tweezers to obtain force-extension curves. The differences in the force-extension curves for the same nanospring under various buffer conditions indicate differential structural integrity of nanosprings. In a 10 mM Tris buffer with 100 mM KCl (pH 7.4), the nanosprings formed a coiled structure owing to the formation of G-quadruplexes in the bridge strands between piers. When stretched up to 40 pN, such G-quadruplexes unfolded whereas relaxation in tension caused unfolded structures to refold, thereby showing a large hysteresis between stretching and relaxing curves (Fig. 2b). However, when the

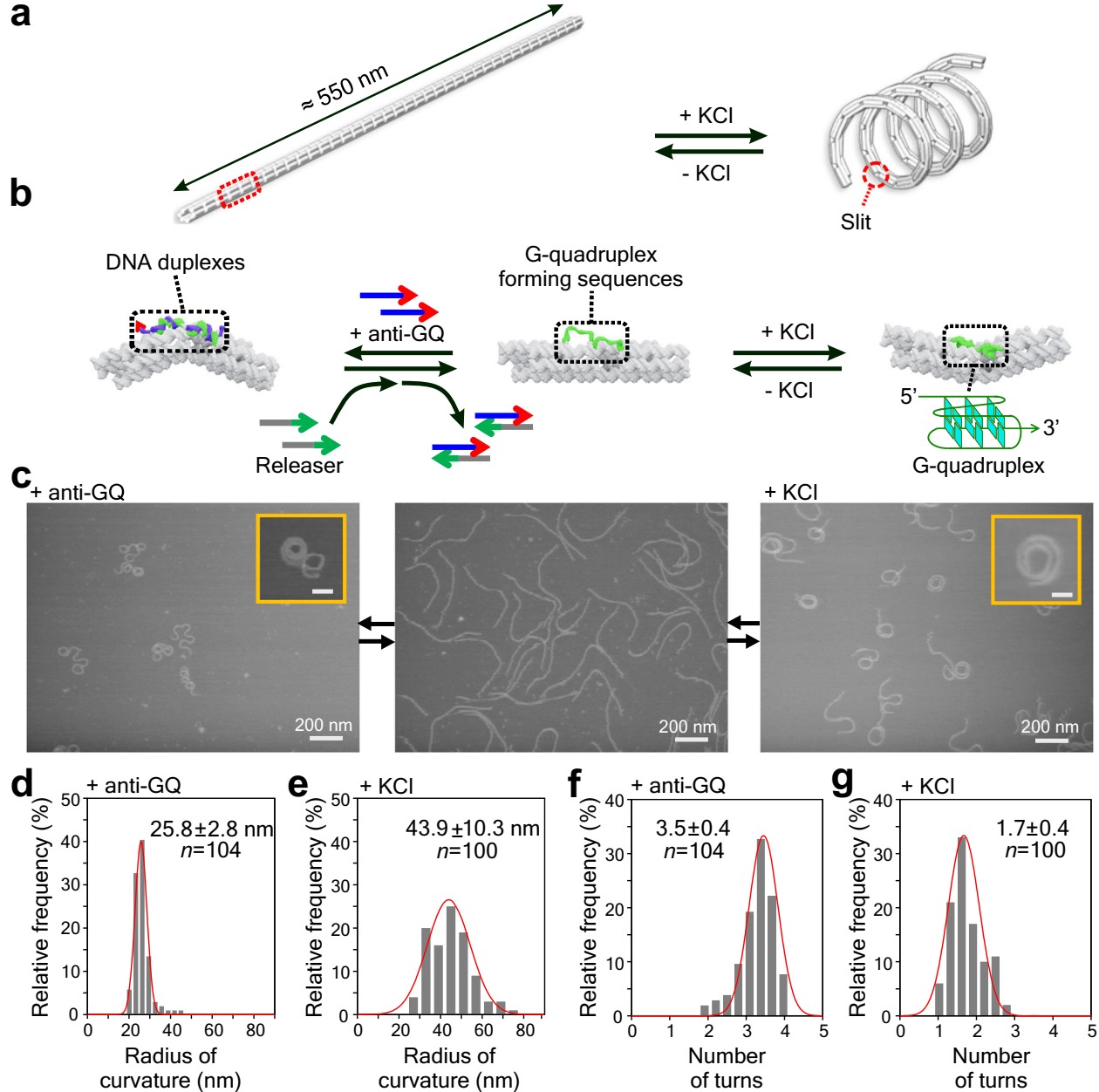

**Fig. 1 | Design of a dual-switching nanospring. a** Reversible transformation of a DNA origami bundle into a spring shape through the cumulative actuation of K[+]-responsive modules. The details of the module (dashed red box) are shown in **b** and Supplementary Fig. 2. **b** Schematics of the module. The ssDNA bridge strand containing a G-rich sequence (5′-GGGTTAGGGTTAGGGTTAGGG-3′) flanked with staple sequences is incorporated in the module. The strand forms a G-quadruplex in the presence of K[+], which leads to the bending of the module. The strand can also hybridize with an anti-GQ strand carrying a toehold sequence (5′-CCCTAACCC-TAACCCTAACCCAGAGAACT-3′). The 21 bp duplex induces bending whose direction is opposite to that induced by the GQ-formation. The anti-GQ strand can be displaced from the module via the toehold-mediated strand displacement process with a releaser strand, 5′-AGTTCTCTGGGTTAGGGTTAGGGTTAGGG-3′, whose

sequence is fully complementary to the anti-GQ strand. **c** Representative AFM images of the nanosprings of over three independent experiments taken after the hybridization with the anti-GQ strand, in the absence of both the anti-GQ strand and 100 mM KCl, and in the presence of 100 mM KCl. Inset scale bar is 50 nm. **d**, **e** Histograms of the curvature radius of the nanospring after hybridization with anti-GQ strands **d** and that in the presence of 100 mM KCl without bound anti-GQ **e**. $n$ represents the total number of nanospring molecules evaluated. **f**, **g** Number of turns calculated from the curvature radius and number of turns for nanospring measured by AFM after hybridization with anti-GQ strands **f** and that in presence of 100 mM KCl without bound anti-GQ **g**. The errors refer to standard deviations (SD). Source data are provided as a Source Data file.

---

same nanospring was introduced into the channel containing 10 mM Tris buffer with 100 mM LiCl (pH 7.4), formation of G-quadruplex was not facilitated, which led to uncoiled nanosprings, resembling a straight topology of DNA bundles. The force-extension curves at this regime show little hysteresis (Fig. 2b insets, "Uncoiled NS"), which confirmed no formation or dissolution of G-quadruplexes between

adjacent piers. Finally, when the nanospring was introduced to the channel that contained anti-GQ oligo (1 μM) in a 10 mM Tris buffer with 100 mM LiCl (pH 7.4), we observed a hysteresis whose size stays between GQ-NS and Uncoiled-NS. This can be explained as hybridization of the anti-GQ oligo onto the bridge strand is faster than the refolding of GQ in the GQ-NS, but slower than the conformational

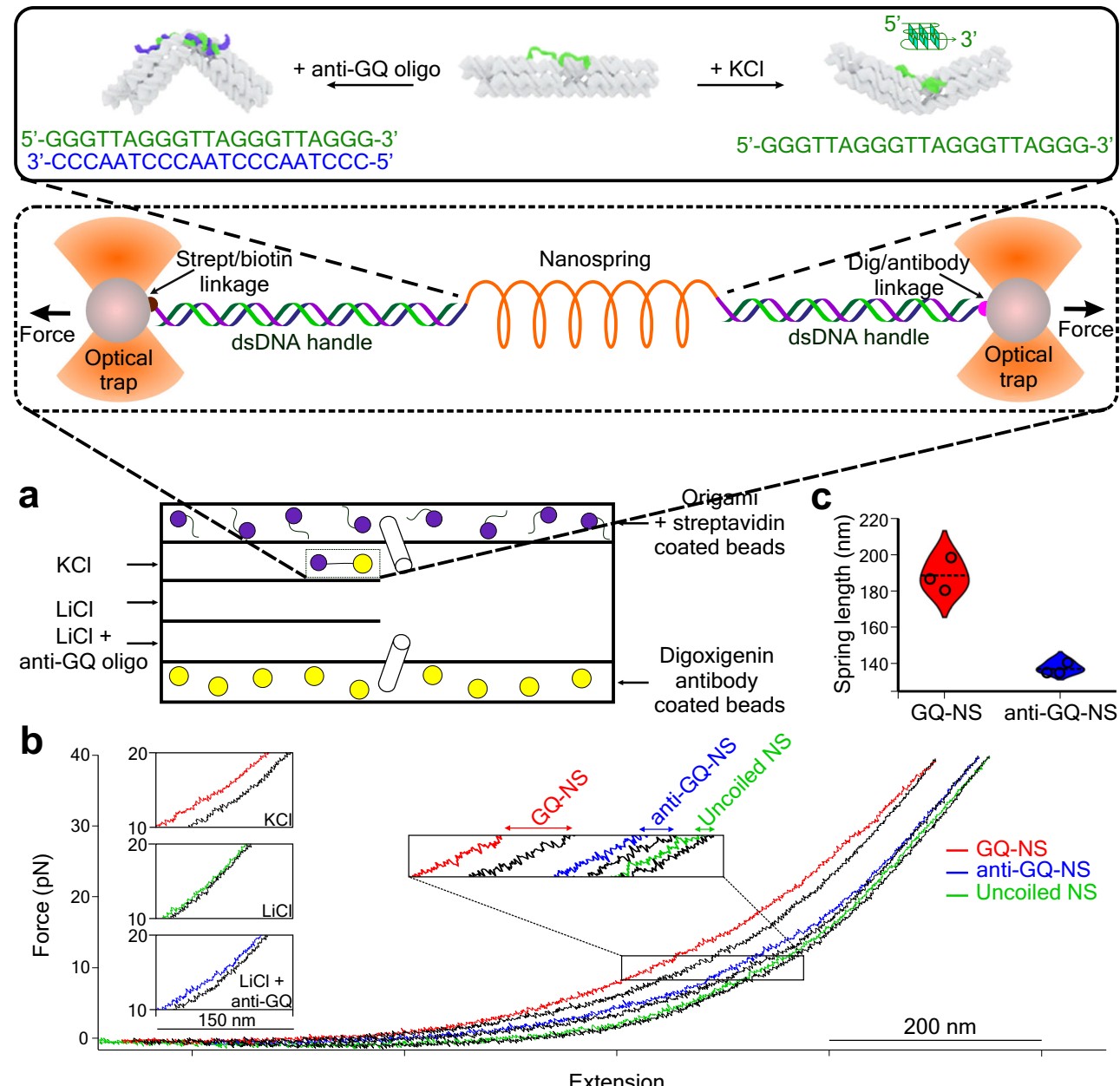

**Fig. 2 | Mechanical properties of nanosprings revealed by force ramping experiments. a** Schematic of a 5-channel microfluidic chamber for force ramping and force-jump experiments of nanosprings. Inset (dotted black box) shows the nanospring tethered between two dsDNA handles attached to beads via strepta-vidin/biotin and digoxigenin/antibody linkages. Beads are trapped in optical tweezers set-up. Inset (solid black box) shows different conformations of nanosprings in different buffers. **b** Force-extension curves from the same nano-spring molecule switched between different channels with insets showing different extent of hysteresis at the range of 10 to 20 pN. **c** Violin plots indicate the length of each nanospring estimated at zero force. $n = 3$ molecules each for GQ-NS and anti-GQ-NS. Each circle represents a data point while dotted lines represent the mean value. Source data are provided as a Source Data file.

fluctuation of the bridges in the Uncoiled-NS. These results clearly indicate reversible topological switch of the nanosprings in 3D space under different conditions.

## Table 1 | Fitting parameters of the nanosprings

| Parameter (unit) | GQ-NS | anti-GQ-NS | Theoretical values |
|---|---|---|---|
| $L_p$ for handles (nm) | 50 | 49 | ≈50 |
| $L_O$ for handles (nm) | 1244 | 1211 | ≈1400 |
| $K_O$ for handles (pN) | 1298 | 1446 | ≈1000–1500 |
| $k$ for nanospring (pN nm$^{-1}$) | 0.04 | 0.03 | N/A |
| $x_O$ for nanospring (nm) | 189 | 138 | N/A |

Next, we estimated the length of nanosprings in different conditions using the extensible Worm-Like Chain (WLC) model complemented by the Hooke's law expression[23]. In the 100 mM KCl buffer, the G-quadruplex containing nanosprings (GQ-NS) had a length of $189 \pm 9$ nm at zero force while in the 100 mM LiCl buffer containing 1 µM anti-GQ oligo, the nanospring (anti-GQ-NS) showed a length of $138 \pm 3$ nm (see Supplementary Fig. 7 and Table 1). Since the numbers of coils in the two nanosprings are $1.7 \pm 0.4$ and $3.5 \pm 0.4$ for GQ-NS and anti-GQ-NS, respectively (Fig. 1f, g), pitch lengths for these nanosprings were estimated as $189 \pm 9$ nm per 1.7 turns $= 110 \pm 30$ nm and $138 \pm 3$ nm per 3.5 turns $= 39 \pm 5$ nm in 3D space.

These structural features (shorter nanospring length and shorter pitches for anti-GQ-NS) at the resting state suggest a

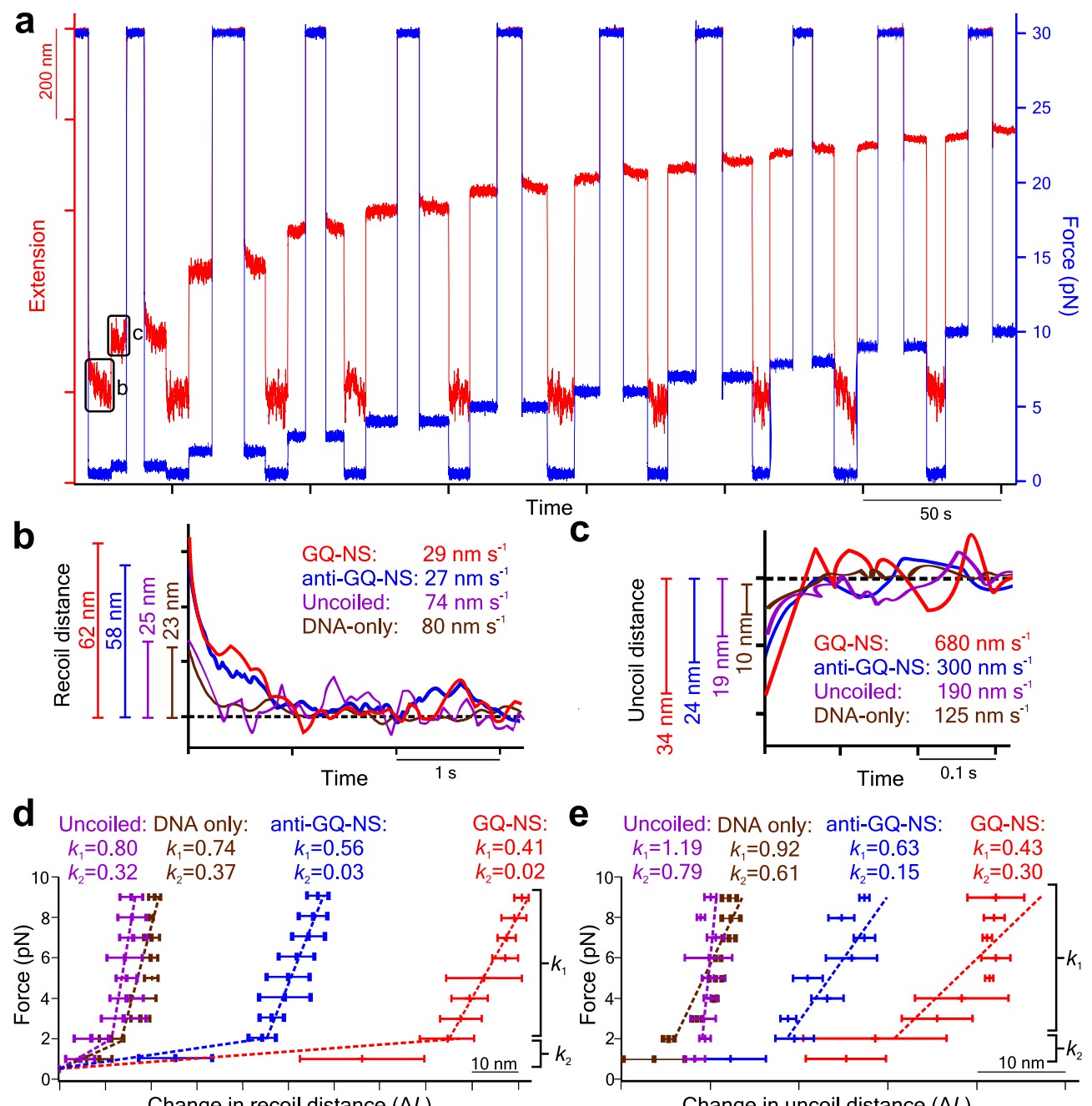

**Fig. 3 | Spring constant measurements of nanosprings by force jump experiments. a** Temporal trace of forces and extensions during different force-jump events of a GQ-NS. A similar typical temporal trace of an anti-GQ-NS is presented in Supplementary Fig. 8. Two boxes marked as b and c in the traces represent recoiling and uncoiling events respectively. Magnified images for the **b** recoiling events of each nanosprings from 30 pN to 0.5 pN and **c** the uncoiling events from 0.5 to 1 pN. Spring constants for different nanosprings were calculated via Hooke's Law ($F/\Delta L$) from **d** recoiling events and **e** uncoiling events. The errors refer to SD for $n = 3$ molecules. Source data are provided as a Source Data file.

stronger spring constant for the anti-GQ-NS vs GQ-NS. To verify this prediction, we measured spring constants for both nanosprings using force-jump methods established recently[23]. We measured the spring constants under either recoiling or uncoiling condition. To measure recoiling spring constants, the nanospring was maintained in a fully stretched state at 30 pN, followed by sudden decrease of tension ranging from 0.5 to 10 pN. For uncoiling, the same nanospring initially maintained at 0.5 pN was suddenly stretched to a high force ranging from 1 to 10 pN (Fig. 3a). As soon as the final force was reached, the extension of the nanospring was monitored over time (Fig. 3b, c).

From these temporal traces, the recoiling and uncoiling kinetics for different nanosprings were calculated (Fig. 3b, c). The GQ-NS and anti-GQ-NS showed similarly slower recoiling kinetics (29 nm s⁻¹ and 27 nm s⁻¹ for GQ-NS and anti-GQ-NS, respectively) compared to the uncoiled nanospring (100 mM LiCl buffer without anti-GQ oligo, 74 nm s⁻¹) or dsDNA construct (89 nm s⁻¹, see Methods for detailed preparation). The slower recoiling rates in nanosprings reflect the sluggish formation of the mesoscale coiling conformation. The longer recoiling distances for GQ-NS and anti-GQ-NS with respect to those observed in the uncoiled nanospring or dsDNA construct indicate mesoscale conformations have more dynamic ranges. In

the duplex DNA or the uncoiled nanospring, the higher-order mesoscale spring conformation is absent, therefore, the recoiling kinetics reflects inherent elastic behavior of the duplex DNA strand or bundles of dsDNA with force, respectively.

From Fig. 3d, e, we retrieved the spring constants of different nanosprings by calculating the slopes of the change in the recoiling distance ($\Delta L$) vs force according to the Hooke's law. We observed that the slopes were located within two regions bifurcated at ≈2 pN, which is attributed to higher entropic contribution for coiling in the low force region (below 2 pN) and higher enthalpic contribution of the coiling above 2 pN[23,26,27]. In both regions for either uncoiling or recoiling transition, we found that the spring constants of GQ-NS and anti-GQ-NS are smaller than the uncoiled nanospring or dsDNA (Fig. 3d, e). This indicates that the nanospring is much softer than the origami backbone itself or duplex DNA strands. It is significant that anti-GQ-NS has higher spring constants than the GQ-NS, which agrees with the observation that anti-GQ-NS has smaller coil radius while shorter overall and pitch lengths at zero force. Close inspection on the $x$-axes of Fig. 3d, e revealed that compared to the uncoiled nanospring or the dsDNA strand, the changes in the uncoiling or recoiling distances at any particular force are significantly greater for the anti-GQ-NS and GQ-NS nanosprings. These observations are consistent with the formation of the mesoscale nanospring topology in 3D space for the anti-GQ-NS and GQ-NS constructs.

As observed from Fig. 3b, c, the recoiling events (time to reach recoiling equilibrium) are much longer than uncoiling events, hence, it is more reliable to retrieve the spring constants in the recoiling, instead of uncoiling force jumps. We also measured spring constants by fitting force-extension curves with an equation[23] combining the worm-like chain model with the Hooke's law (see Methods). We found that the spring constant values (0.04 and 0.03 pN nm$^{-1}$ for GQ-NS and anti-GQ-NS, respectively, when fitted within the force range 0.1 to 8.2 pN) are located between those obtained from the recoiling force-jump experiments at <2 pN (0.02 and 0.03 pN nm$^{-1}$ for the GQ-NS and anti-GQ-NS, respectively) and those at 2 to 10 pN (0.41 and 0.56 pN nm$^{-1}$ for the GQ-NS and anti-GQ-NS, respectively). Both methods, therefore, validated the spring constant measurements. However, the force-jump experiment is more accurate as it can precisely evaluate the recoiling and uncoiling events at a particular force. In contrast, force-extension fitting assumes the spring constant does not vary in the fitted force range, which may not be true as conformation of soft DNA nanosprings is expected to change with force. In addition, in the slow and continuous force extension curves, the effective mechanical quantifications of the whole construct (nanospring + DNA handles, see Fig. 2a) are mainly contributed from long duplex DNA handles (≈2.3 μm), instead of the GQ or the GQ-antiGQ duplex (<10 nm) formed in each junction of the neighboring piers in the nanospring. On the other hand, force jump assay provides a different approach to effectively differentiate the mechanical quantifications (such as spring constants) between the dsDNA handles and the DNA nanospring, which are respectively based on the transition kinetics in the DNA handles and GQ or GQ-antiGQ duplex in nanospring junctions in their responses to rapid force variations (see Fig. 3b, c).

It is noteworthy that nanosprings can maintain their structural integrity in the force range we have applied (up to 30 pN). First, consecutive force-extension curves revealed overlapping stretching and relaxing traces in the same nanospring construct (Supplementary Fig. 9). Second, consecutive force jump experiments also displayed similar uncoiling and recoiling events for the same nanospring (Supplementary Fig. 10). Both experiments indicated intact nanospring structures under 30 pN force.

## Higher-order mesoscale topology of nanosprings

To obtain insights into the higher-order structures of nanosprings, we performed high-resolution AFM imaging, which revealed a series of "slits" along the DNA bundle corresponding to the repeated modules (Figs. 1a and 4a, b). It is noteworthy that slits were observed inside of the curves on GQ-NS but outside of the curves on anti-GQ-NS, indicating that the backbone bending directions are opposite as expected (Supplementary Figs. 11 and 12). We reasoned that such a difference in the backbone orientation may result in different chirality (i.e., left- or right-handedness) in the nanospring helices.

To test this hypothesis, next, we performed high-resolution AFM imaging on the nanosprings having intra-structure crossing-over points. Since AFM is a surface topography imaging technique, sectional profile analysis at a crossing-over point of such a structure should allow us to judge which bundle is lying underneath (Fig. 4c), which in turn enables us to distinguish the handedness of the nanospring. Sectional lines were taken in such a way that the DNA bundle along with the line A-B is underneath the other when the helical structure is right-handed, whereas the DNA bundle along with the line C-D is underneath the other when the structure is left-handed (Fig. 4d–i). To determine such spatial features, we surveyed up-hill regions because high-speed AFM cannot accurately image down-hill regions due to parachuting of the AFM tip[28]. Uphill slope values at the crossing-over point (slope$_{AB}$ and slope$_{CD}$) were then compared to calculate change in slopes (=slope$_{AB}$–slope$_{CD}$), a positive value of which indicates a right-handed structure. In all our measurements, change in slopes for anti-GQ-NS and GQ-NS both showed positive values, suggesting both nanosprings are right-handed (Fig. 4j). When we varied the scan directions of the AFM tip, we also observed the same chirality in both nanosprings (Supplementary Fig. 13). Given that GQ-NS and anti-GQ-NS shared the right-handed helix chirality, the opposite backbone orientations observed above (Fig. 4a, b) therefore suggest that achiral backbone orientation stemmed from the linear chemo-mechanical force is not responsible for the higher-order mesoscale chirality. Instead, the slightly overwinding helicity in the B-DNA based backbones may determine the nanospring chirality[15].

To confirm these intriguing higher-order mesoscale structures of DNA nanosprings, we performed coarse-grained molecular dynamics simulations[29–32]. To relax the initial straight conformation into coiled structure, a simulation of relatively long time is necessary. Because a full-size nanospring with 37 units of bending modules cannot be simulated within a practical time, we built initial configurations of substructures with 13 units and simulated them for 3 μs. Moreover, since the force field of oxDNA is optimized for canonical B-form DNA, it cannot simulate G-quadruplex structure. Instead of G-quadruplex structure, we used 4 nt poly-T linker (4 T) given that both have similar end-to-end distances. Here, we assume that the distance between the ends of G-quadruplex structure is about 2 to 3 nm[33] and that between neighboring bases of single-stranded DNA is 0.68 nm[34]. For the purpose of comparison, we simulated the structures of the 21 bp and the 4 nt linkers (Supplementary Figs. 14 and 15).

From the plot of root mean square deviation (RMSD), we assumed that the structures were fully relaxed after 2 μs of simulation and decided to use the result of last 1 μs for further analysis (Fig. 5a). Using 100 snapshots with 10 ns time interval of the last 1 μs, we computed the root mean square fluctuation (RMSF) and found that the structures were subject to large thermal fluctuation, indicating that the nanospring is a soft material (Fig. 5b).

For the coiling arrangement, however, backbones appeared inside and outside for the cases of the 21 bp and 4 nt linker, respectively, which are consistent with experimental observations (Fig. 4a, b). To quantify the difference, the simulation results were fitted to a formula of helix (Supplementary Fig. 16). The difference in average radius indicates that inside and outside of the structure were opposite in the

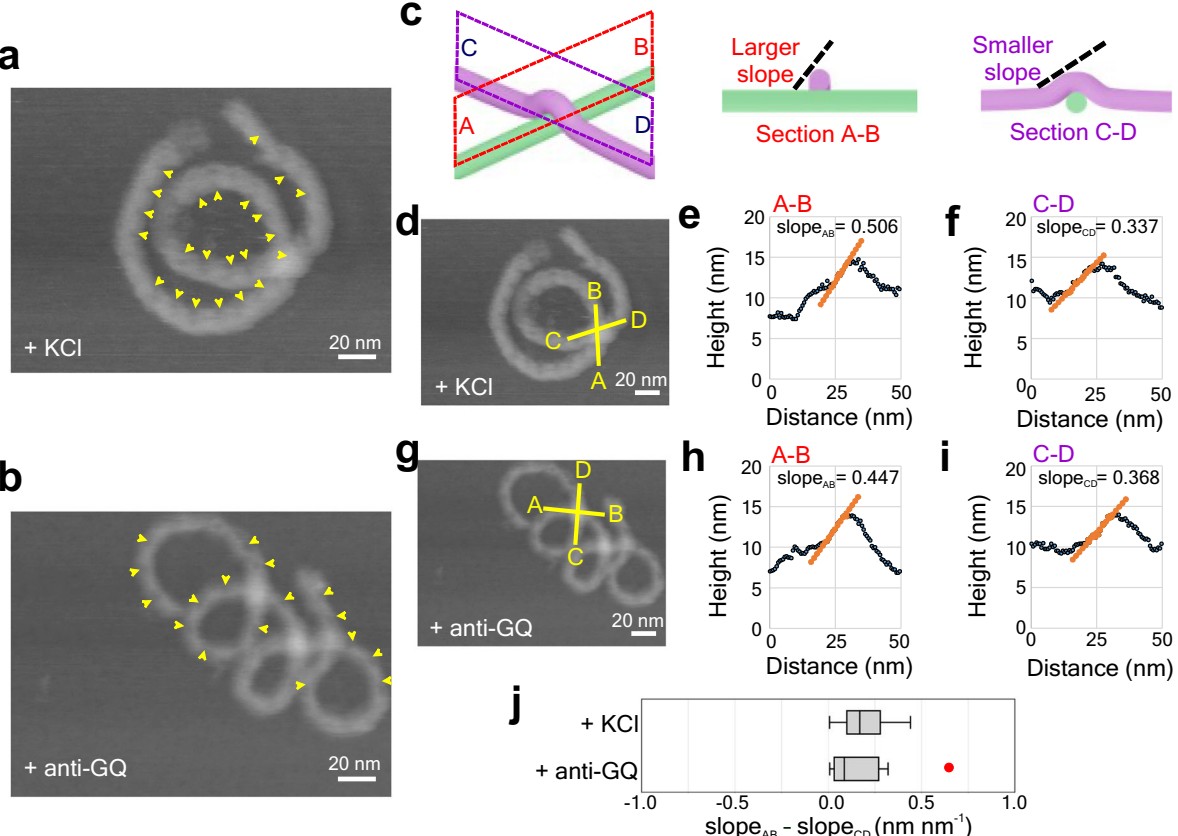

**Fig. 4 | Higher-order topology of nanosprings revealed by AFM. a, b** High-resolution AFM images of two typical nanosprings in presence of **a** 100 mM KCl for $n = 9$ molecules examined over three independent experiments and **b** 10 μM anti-GQ strands for $n = 7$ molecules examined over four independent experiments. The yellow arrows mark the bridging regions (slits) between two piers (see Fig. 1a, b). The backbones of the nanosprings are inside out between images in **a** and **b**. More images are shown in Supplementary Figs. 11 and 12. **c** Schematics of two crossing bundles on a 2D surface and section views along two orthogonal planes. A larger slope indicates that the bundle along the sectional plane (A-B) is lying underneath. **d** High-resolution AFM image of the nanospring in the presence of 100 mM KCl. **e, f** Slope analyses along the two directions representing two crossing bundles in the presence of 100 mM KCl. **g** High-resolution AFM image of the nanospring in the

presence of 10 μM anti-GQ strands. **h, i** Slope analyses along the two directions representing two crossing bundles in the presence of 10 μM anti-GQ strands. Note that sectional lines were taken in such a way that the DNA bundle along the line A-B is underneath the other when the structure is right-handed, whereas the DNA bundle along the line C-D is underneath the other when the structure is left-handed. **j** Box plots of the change in slopes (=slope$_{AB}$–slope$_{CD}$). A positive value indicates the right-handed structure. The boxes represent Inter Quarter Range (25th–75th percentiles), the center line indicates the median, and the whiskers extend to the maximum and minimum values ($n = 9$ molecules examined over three independent experiments for +KCl; $n = 7$ molecules examined over four independent experiments for +anti-GQ). Red dot is an outlier point and excluded from the analysis. Source data are provided as a Source Data file.

21 bp and 4 nt linker cases (Fig. 5c, d). In the case of the 21 bp linker, over-extended bridges push the piers apart, resulting in the inside backbones. On the other hand, tension from the short 4 nt linker pulls the piers together to make a coil with backbones outside. Finally, the tendency of the difference in radius that the 4 nt has bigger coils than the 21 bp linker structure (due to the flipping backbone topologies of the nanosprings in the 4 nt and the 21 bp linker structures, the backbone radius of one topology should be compared with that of the pier topology from another structure) agrees with the experiment. From the analysis, we also found that the thermal fluctuations of the edge parts of the structure with the 21 bp linker had a non-negligible impact on the radius and pitch of the helix in some frames (Outlier points in Fig. 5d, Supplementary Fig. S17). Nevertheless, given the uncertainty of the thermal fluctuations, the 21 bp linker did show a trend of shorter pitch length with respect to the 4 nt linker nanospring, which was again consistent with that calculated from the AFM and optical tweezers' measurements (Figs. 1f, g and 2c).

In both the 21 bp and 4 nt linkers, the chiralities of DNA nanospring were right-handed for the last 1 μs (the final snapshot in Fig. 5c), which fully agrees with the analysis of AFM imaging. On the other hand, when the bridges were removed or replaced with single-stranded 21 nt

linkers, the helix shapes were not shown, and rather straight or arched structures were formed after the simulation (Supplementary Figs. 18 and 19).

## Estimation of chemo-mechanical force in the mesoscale topologies

The mechanical stability of human telomeric G-quadruplex used in this study is about 20 pN[35]. Given there are two G-quadruplexes in each junction (Supplementary Fig. 1), we estimated 40 pN is sufficient to bend the two adjacent origami piers in an arch through which nanospring coils. On the other hand, the stiffness of duplex DNA is strong enough to bend the neighboring DNA origami piers to form a nanospring with the opposite backbone orientation. The force in a duplex DNA can be estimated as 31 pN by assuming its end-to-end distance equivalent to the contour length of the dsDNA with 50 nm persistence length[25], which gives ≈62 pN to push the piers apart as there are two duplex strands in each junction (Supplementary Fig. 1). It is significant that a gentle linear force in the range of 40 to 60 pN is sufficient to control higher-order 3D achiral topology of mesoscale DNA assemblies. With less than 1 nm persistence length[36], polypeptides are softer than dsDNA. Therefore, it

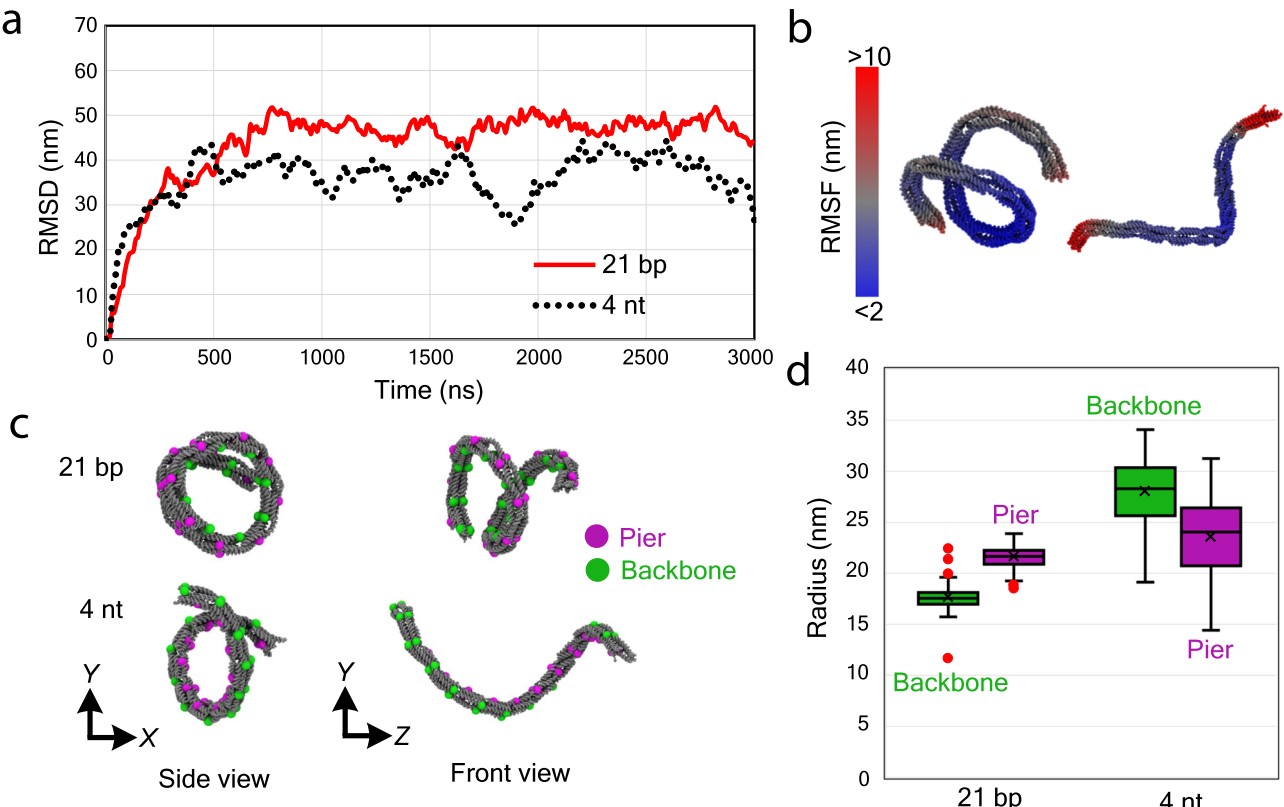

**Fig. 5 | Topology of nanosprings revealed by coarse-grained simulation.**
**a** Trajectory of root mean square deviation (RMSD) as an index for simulation equilibration. To compute the RMSD, the squared distance from the initial position at 0 ns was computed for each entity at each time step, and root mean of the values were plotted. Horizontal and vertical axes are time and RMSD, respectively. Red solid and black dotted lines correspond to the 21 bp and 4 nt linker structures. **b** Visualization of root mean square fluctuation (RMSF) of the last 1 μs as an index of thermal fluctuation (21 bp linker on the left; 4 nt linker on the right). To compute the RMSF, the squared distances from reference position were computed for a total of 100 frames of simulation from 2000 to 3000 ns for each entity. The reference position for each entity was defined as the averaged coordinate of the 100 frames.

Each entity of oxDNA at the last frame of simulation is colored by the RMSF values ranging from 2 to 10 nm (outlier values smaller or larger than 2 or 10 nm thresholds have the same colors of 2 or 10 nm, respectively). **c** Snapshot of the structures at 3 μs of the simulation. The entities of backbone and pier used for the computation are emphasized in green and magenta spheres. **d** Differences of radius of fitted helix between backbone and pier, and between the 21 bp and 4 nt linker structures (n = 100 for each measurement). In the graph, red dots are outlier points which are excluded from analysis, boxes represent the first and third quartiles, middle line shows the median, cross point is the mean, and the whiskers show minimum and maximum values. Source data are provided as a Source Data file.

is anticipated that even smaller force may be required to modulate higher-order topology of mesoscale protein complexes. Given that tens of pN force is routinely present inside cells, which can be generated by motor proteins such as helicase and polymerases[37] or by formation/unfolding of chemical structures, we expect that mechanical modulation of higher-order biological mesoscale structures is prevalent inside cells.

In summary, we have used DNA origami nanospring as a model system to elucidate higher-order conformation dynamics of mesoscale biomolecular assemblies. We have observed that duplex and quadruplex DNA formed in the junction of two adjacent origami modules have rather different effects on the structure and property of DNA nanosprings, resulting in different coil radii, spring/pitch lengths, and spring constants. While these two DNA secondary structures control the reversible orientational flipping of the nanospring backbones, the linear mechanical forces associated with the formation of these two structures do not change the right-handed helical chirality of the nanosprings. This result indicates that chirality in higher-order helices should be determined by rotational torques inherent in the nanospring backbones or junctions. From the mechanical stability of these two secondary structures, we conclude that linear chemo-mechanical force of ≈40 to 60 pN, which can be generated via formation/unfolding of chemical structures, is sufficient to control achiral

higher-order mesoscale structures such as DNA nanospring assemblies. We anticipate these results not only provide insights on the origin of chiral helicity in mesoscale assemblies, but also lead to new, topologically based caging/uncaging strategies that are reversible in solutions.

## Methods

### Materials

Scaffold DNA (p8064) used for the synthesis of DNA origami was purchased from Tilibit Nanosystems (Garching, Germany) while other DNAs used in the preparation such as staple and bridge strands were purchased from Eurofins Genomics Tokyo (Tokyo, Japan). The pET-26b (+) plasmid used as PCR template was acquired from Novagen (Darmstadt, Germany) and the required PCR primers were obtained from Japan Bio Service (Saitama, Japan). Restriction enzymes were purchased from New England Biolabs (Ipswich, MA, USA). Chemicals such as KCl (99.0–100.5%), MgCl$_2$ (≥99.9%), and EDTA (99.4–100.6%) were obtained from VWR. The polystyrene beads coated with either anti-digoxigenin or streptavidin were made available from Spherotech (Lake Forest, IL, USA).

### DNA origami nanosprings

The initial model of DNA origami nanospring was conceptualized via caDNAno software[8] for strand routing and CanDo[38,39] for the structure

prediction. The entire structure assembly of the origami nanospring was carried out by mixing 10 nM circular single-stranded scaffold DNA (p8064) with ≈40 nM staple strands, along with bridge strands in 40 µL of the folding buffer containing 5 mM Tris·HCl (pH 8.0), 1 mM EDTA, and 15 mM MgCl$_2$. Next, the mixture solution was incubated at 65 °C for 15 min, and then annealing of complementary DNA sequences was facilitated by reducing the temperature from 60 to 45 °C at a rate of −1.0 °C h$^{-1}$. The obtained assembled mixture was purified using PEG-precipitation[40]. In that process, the annealed mixture was mixed with a precipitation buffer (15% PEG 8000 (w/v), 5 mM Tris·HCl (pH 8.0), 1 mM EDTA, and 505 mM NaCl) in the volume ratio 1:1 and then centrifuged at 16,000 × $g$ for 25 min. Finally, the supernatant was removed, and the pellet was dissolved in the buffer with a designated concentration of KCl (0 or 100 mM) for experimental use.

## Preparation of the poly-digoxigenin and biotin labeled dsDNA handles

The poly-digoxigenin and the biotin labeled dsDNA were prepared using an established protocol[23]. In brief, the two 2520-bp dsDNA handles were synthesized via PCR amplification of pET-26b (+) plasmid. For that, the forward primer was comprised of "5′-Staple sequence-O-(CH$_2$)$_2$-O-(CH$_2$)$_2$-O-Primer sequence". The staple sequence provided a single-stranded overhang on one end of each handle which hybridized with either end of a distinct staple present in the nanospring origami. The other end of each handle was labeled with either biotin or digoxigenin. To synthesize the biotin labeled handle, the reverse primer was directly modified with 5′ biotin while in case of digoxigenin labeled handle, the PCR amplified product was cleaved with RPSacI and further labeled with poly-digoxigenin-dUTPs using terminal transferase (TdT) enzyme. Biotin/streptavidin and digoxigenin/anti-digoxigenin linkages have been extensively used for decades in force-based single molecule assays because of their high specificity, binding affinity and force stability[41]. Although single digoxigenin/anti-digoxigenin interaction has lower force stability (≈25 pN) compared to biotin/streptavidin interaction (≈200 pN), multiple digoxigenin to anti-digoxigenin interactions significantly increase the mechanical stability. Moreover, use of different linkers at the end of DNA handles increases the tethering yield of individual nanospring constructs to different polystyrene beads for single-molecule assays.

Primers for the digoxigenin labeled handle
Forward primer:
5′- TTT AAA GGG CAG TGT TGT TCC AGT TTG CCC GAG ATA GGG TTG GAA AAA CCG TCT ATC A -X- CGC CGA TCA ACT GGG TGC CAG CGT
Reverse primer:
5′- AAA AAA AAG AGC TCG GGT TCG TGC ACA CAG CCC AGC TT
Primers for the biotin labeled handle
Forward primer:
5′- TTT CAT AGT TAC TGA GTT TCG TCA CCA CCC ATG TAC CGT AAC AGC GTA ACG ATC TAA AGT TTT GTC-X- CGC CGA TCA ACT GGG TGC CAG CGT
Reverse primer:
5′-[Biotin]-GGGTTCGTGCACACAGCCCAGCTT
X = O-(CH$_2$)$_2$-O-(CH$_2$)$_2$-O

## Preparation of handle-conjugated DNA origami nanosprings

For the synthesis of origami nanosprings with dsDNA handles, 10 nM scaffold DNA (p8064), ≈40 nM staple strands, bridge strands, and 10 nM each of two DNA handles (biotin-handle and poly-DIG handle), were mixed in 60 µL of folding buffer comprised of 5 mM Tris·HCl (pH 8.0), 1 mM EDTA, and 15 mM MgCl$_2$. This mixture was then incubated at 65 °C for 15 min, and then annealed by decreasing the temperature from 60 to 45 °C at a rate of −1.0 °C h$^{-1}$. The assembled structure was then purified by PEG-precipitation as described in the above section.

## Preparation of a control construct without DNA origami nanosprings

The control construct without DNA origami was prepared by sandwiching a DNA sequence, 5′-CTAGACGGTGTGAAATACCGCACA-GATGCGTTGAACTATACAACCTACTACCTCATTTTTGAGGTAGTAGG TTATCGCCAGCAAGACGTAGCCCAGCGCGTC-3′ between two dsDNA handles: 2028-bp dsDNA and 2690-bp dsDNA[23]. The 2028-bp dsDNA handle was synthesized from PCR of pBR322 plasmid while the 2690-bp dsDNA handle was prepared by the restriction enzyme digestion of pEGFP plasmid using EagI and XbaI endonucleases. The whole construct was prepared by first annealing the phosphorylated DNA sequence with two DNA oligonucleotides: 5′-CGCATCTGTGCGG-TATTTCACACCGT-3′ and phosphorylated 5′-GGCCGACGCGCTGGGC-TACGTCTTGCTGGC-3′, starting the annealing process at 95 °C for 15 min and then reducing to 20 °C at the rate of −1.0 °C min$^{-1}$. This annealed product was then ligated to biotin labeled 2028-bp dsDNA handle. Next, the agarose gel purified product was finally ligated to poly-digoxigenin labeled 2690-bp dsDNA handle which was eventually used for the force-jump assays in optical tweezers set-up.

## Single-molecule force ramping assays

The single-molecule force ramping assays were carried out in an optical tweezers-setup. For that, the synthesized nanosprings with 2520-bp dsDNA handles (containing poly-digoxigenin on one end while biotin on another end) were diluted to ≈2 ng. Next, the whole construct was then incubated with 0.1% solution of streptavidin coated polystyrene beads for 10 min at room temperature, resulting in the formation of biotin-streptavidin complex. This sample was further diluted in 1 mL of 100 mM Tris-KCl buffer (pH 7.4) along with 15 mM MgCl$_2$ and 1 mM EDTA. The sample solution was then injected from the top channel of the 5-channel microfluidic chamber (shown in Fig. 2a). Likewise, the lowermost channel was flown with anti-digoxigenin antibody coated polystyrene beads. The middle channels were then flown through appropriate buffers containing 100 mM KCl (the 2nd channel from top), 100 mM LiCl (middle channel) and 1 µM anti-GQ oligonucleotide (5′-CCCTAACCCTAACCCTAACCC) in 100 mM LiCl (the 4th channel from top) to imitate the conditions favorable for the formation of GQ-nanospring (GQ-NS), uncoiled nanospring and anti-GQ-nanospring (anti-GQ-NS), respectively. Top and bottom channels were linked to other channels via 0.025 mm ID capillary tubes.

For the assay, an anti-digoxigenin coated bead from the lowermost channel was trapped first and moved to the channel with Tris-KCl buffer. Then another bead attached with origami flowing through capillary from the topmost channel was trapped. A successful sandwiching of the nanospring occurred between two beads due to dig/anti-dig antibody interaction on one end and biotin/streptavidin interaction on another end of dsDNA handles as shown in Fig. 2a. Next, force ramping assays were performed with a loading rate of ≈5.5 pN s$^{-1}$ in the 2nd channel (from top) reaching a maximum force of 40 pN. The force-extension curve in KCl region was recorded and then, the beads containing the same molecule were moved to the middle channel (containing LiCl) where other sets of force-extension curves were recorded. Finally, the same molecule was brought to the 4th channel from top (containing anti-GQ oligos in 100 mM LiCl) and again several force-extension curves were recorded.

## Force-jump experiments

A 3-channel microfluidic setup was prepared for the force-jump experiments. For GQ-NS, the 10 mM Tris-buffer condition was maintained with 100 mM KCl while for anti-GQ-NS, it was maintained with 100 mM LiCl. The tethered nanospring construct (as done in force-ramping experiment) was fully stretched and maintained at 30 pN, which was followed by a sudden reduction to a force of 0.5 pN within 10 ms (Fig. 3a). The data showing the changes in force, recoiling distance, and time were recorded. Similarly, other force jump transitions

were performed ranging from initial high force of 30 pN to the low forces of 1, 2, 3, to 10 pN respectively. To perform the uncoiling events, the tethered DNA nanospring was maintained at 0.5 pN and then stretched suddenly to a force of 1 pN and data showing the changes in force, uncoiling distance, and time were recorded. Similarly, other force jump transitions were performed ranging from initial low force of 0.5 pN to the high forces of 1, 2, 3, to 10 pN respectively. Similar force jump events were carried out for the anti-GQ-NS recording the changes in force, recoiling distance, uncoiling distance, and time.

## Fitting model for the force-extension curves

In the low force region ($F = 0$ to 10 pN), force-extension curve of a nanospring can be described by the Hooke's law,

$$F = k(x + x_0) \tag{1}$$

where $k$ and $x_0$ are the spring constant and initial spring length ($F = 0$ pN), respectively. Force-extension curve of the dsDNA handles can be described by the Worm-like Chain (WLC) model[42],

$$F = \left(\frac{k_B T}{L_p}\right)\left[\frac{1}{4\left(1 - \frac{x_0}{L} + \frac{F}{K_0}\right)^2} - \frac{1}{4} + \frac{x}{L_0} - \frac{F}{K_0}\right] \tag{2}$$

where $k_B T$, $L_p$, $L_0$, and $K_0$ are Boltzmann's constant times absolute temperature, persistence length, contour length, and elastic modulus, respectively. Consequently, force-extension curves of the nanospring and dsDNA handles can be fitted by the combination of the Hooke's law and worm-like chain (WLC) model. Since WLC model is an implicit function for force $F$, MATLAB was used to solve the equation to get the explicit function,

$$x = f_{WLC}(F) \tag{3}$$

where $f_{WLC}(F)$ is the numerical formula about parameter $F$ obtained from MATLAB. Then, the equation $x = f_{WLC}(F) + F/k + x_0$ was used to fit force-extension curve in the low force region. For the multivariable fitting, the first step is to hold the dsDNA handle parameters ($L_p$, $L_0$, and $K_0$, or persistent length, contour length, and stretch modulus, respectively) to get the force calibration value, which can eliminate experimental deviation near the zero force. After the force calibration, we fitted the force-extension curves by the equation calibrated by $F_0$,

$$x = f_{WLC}(F - F_0) + \frac{F - F_0}{k} + x_0 \tag{4}$$

with all parameters changing freely to obtain the final fitting results, which are depicted in Table 1 and Supplementary Fig. 3.

## Agarose gel electrophoresis

The samples were loaded for electrophoresis on a 1.0% or 1.5% agarose gel containing 5 mM $MgCl_2$ in a 0.5× TBE (Tris-borate-EDTA) buffer solution (pH 8.3) at 90 V and 4 °C. The gels were then imaged with ChemiDOC MP (Bio-Rad Laboratories, Inc., CA, USA) using SYBR Gold nucleic acid gel stain (Thermo Fisher Scientific, MA, USA) as the staining dye.

## AFM observation

High-speed AFM (HS-AFM) imaging was performed using tip scan high-speed AFM (BIXAM, Olympus, Tokyo, Japan), which was improved based on a developed prototype AFM[43]. A 2 μL drop of the 0.5 to 1 nM sample in buffer composed of 5 mM Tris-HCl (pH 8.0), 15 mM $MgCl_2$, 1 mM EDTA with or without 100 mM KCl was deposited onto a freshly cleaved mica surface (diameter 3.0 mm) and incubated for 1 min. The

surface was subsequently rinsed with 10 μL of the same buffer and then scanned in ≈120 μL of the buffer containing designated concentrations of KCl. Small cantilevers (9 μm long, 2 μm wide, and 100 nm thick) with an electron-beam-deposited carbon tip (tip length ≈2 μm, tip radius <10 nm) having a spring constant of 0.1 N m⁻¹ and a resonant frequency of ≈300–600 kHz in water (USC-F0.8-k0.1-T12; Nanoworld, Neuchâtel, Switzerland) were used to scan the sample surface. The 320 × 240-pixel images were collected at a scan rate of 0.5 frames per second with tapping mode. The images were analyzed using AFM scanning software (Olympus) and ImageJ software (http://imagej.nih.gov/ij/).

## Coarse-grained simulations

**oxDNA simulation.** We used oxDNA coarse-grained model[29] to predict the molecular dynamics of the DNA nanospring because oxDNA has widely used to simulate DNA nanostructures[30,32]. To handle minor and major grooves of DNA and effects of salt, we employed the improved version of the oxDNA model[31]. Topology and initial configuration files for oxDNA simulation were converted from caDNAno-formatted files[8] that were created by a home-made script. The file conversion was done by a TacoxDNA software[44]. For the oxDNA, we downloaded and installed the version 3.4.2 to a Linux computer (Ubuntu 20.04.1). The computer was equipped with AMD Ryzen Threadripper PRO 3955WX 16 cores, 128 GB memory, and GeForce RTX3090 graphic boards. For the purpose of general-purpose computing on graphics processing units, a CUDA driver of v.470.103.01 and CUDA v.11.4 were installed.

**Analysis and visualization of simulation.** Using reported methods[45,46], we defined the oxDNA parameters such as total steps: $3 \times 10^8$, temperature: 0.00938 (298.15 K), salt concentration: 0.5, cutoff radius: 2.0, max backbone force: 5, Verlet skin: 0.05, diffusion coefficient: 2.5, simulation type: MD, interaction type: DNA2, thermostat: John, all of which are written in oxDNA units. Assuming the time unit of oxDNA is 3.03 ps[47], simulations of 3 μs were performed in each computation. The result of oxDNA simulation were visualized by Visual Molecular Dynamics[48] and cogli[49]. Fitting the three-dimensional coordinates to a helix were done by a home-made Scilab script that implemented the HELFIT algorithm[50]. To fit the simulation result, we picked 26 entities of the oxDNA each for backbone and pier, and computed average 13 coordinates along the structures.

## Statistics and reproducibility

No statistical method was used to predetermine sample size. No data were excluded from the analyses. The experiments were not randomized. The investigators were not blinded to allocation during experiments and outcome assessment.

## Reporting summary

Further information on research design is available in the Nature Portfolio Reporting Summary linked to this article.

# Data availability

The data that support the findings of this study are available from the corresponding authors upon request. Source data are provided with this paper.

# Code availability

All custom codes are available from the corresponding authors upon request.

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

## Acknowledgements
H.M. thanks NIH (R01 CA236350 and R01 CA252827) and NSF (CBET1904921) for the financial support. H.M. also thanks the financial support from Lundbeck Foundation grant number R346-2020-1890. This work was also supported by the Japan Society for the Promotion of Science (JSPS) Grant-in-Aid for Scientific Research (KAKENHI; grant numbers 19H04201, 21H05864, 22H03682, and 23H04416 to Y.S. and 22K12255, 21H04434, 20H05971 and 19KK0261 to I.K.). Y.S. and I.K. thank Kotaro Watanabe for the assistance in CG modeling. This publication was made possible in part by support from the Kent State University Open Access Publishing Fund.

## Author contributions
H.M. and Y.S. conceived and supervised the experiments. Y.S. and I.K. contributed for the simulation experiments. D.K. and J.J. performed the mechanical unfolding and force-jump experiments. Y.S. and E.M. performed AFM imaging and analyses. H.M., Y.S., I.K. and D.K. co-wrote the manuscript.

## Competing interests
The authors declare no competing interests.
