## [Peer Review File · Nature Communications]

REVIEWER COMMENTS

Reviewers #1 and #2, who have reviewed jointly (Remarks to the Author):

The authors have described a DNA origami nanospring with two bending mechanisms that can alter the properties of the spring. This improved version of their earlier published nanosprings exhibits a responsive structure and the authors characterize its spring constant using force jump experiments. Interestingly, the nanospring maintains its unique handedness even when its backbone is flipped inside-out which the authors attribute to the inherent rotational torque in DNA. The potential for numerous nanomechanical applications makes this nanospring an exciting discovery. However, there are a few concerns that the authors must address before publication:

The writing of the manuscript included a significant number of grammatical errors and this contributed to making the readability of the manuscript fairly poor. Moreover, the text was dense and at times difficult to follow. So I would highly encourage authors to proof-read the text and attempt to make the writing more concise.

The histogram in figure 1d has a poor fit (indicated in red) that dips below zero near 10 and 40 nm. The data binning in the histogram seems too wide and could be the cause of the poor fit. The authors are encouraged to narrow the bin width and, if necessary, add additional data points from AFM measurements to improve the quality of the fit.

The nanospring turn calculation seems inconsistent in the GQ-NS and anti-GQ-NS nanosprings (estimated below). Further, the authors use a different turn number for the anti-GQ-NS spring in page 8, where it is claimed to be 3.2 turns. The authors must rectify this discrepancy and also report the exact contour length (with error if possible) used in the calculations. The contour length must be supported with citations or measured from existing AFM images.

GQ-NS nanospring: $550\text{nm}/(2 \times 3.14 \times 45.0\text{nm}) = 1.94$ turns (reported as 1.8 ± 0.4).

anti-GQ-NS spring: $550\text{nm}/(2 \times 3.14 \times 26.7\text{nm}) = 3.3$ turns (reported as 3.4 ± 0.4).

In the force jump experiment the authors describe that "For GQ-NS, the 10mM Tris-buffer condition was maintained with 100 mM KCl while for anti-GQ-NS, it was maintained with 100 mM LiCl." This seems to contradict the single-molecule force ramping assay where anti-GQ-NS was stretched in a solution of 1 μM anti-GQ oligos in 100 mM LiCl. If the authors did perform the experiment without an excess of anti-GQ oligos, the authors need to demonstrate that the anti-GQ oligos on the nanosprings do not denature under force. If possible, the authors should further quantify the number of cycles that the nanospring can be stretched and relaxed without compromising its structural properties.

Could the recoiling and uncoiling rate of nanosprings change with KCl or oligo concentration? The authors need to elaborate further on this dependence. How did the authors arrive at the concentrations reported in the experiments?

It unclear if the force jumps performed in figure 3a were done on GQ-NS or anti-GQ-NS. The authors have reported only one of the nanospring temporal traces for the force jumps. The force jump data for the second nanospring could be added to the supplementary information.

In figure S3, lane 4 seems to have a higher molecular weight structure than the initial nanospring as well as other lanes when added with 1:100 fold excess of releaser strand. Is this an artifact or reproducible phenomenon? The authors need to describe the lane components in further detail than it has already been done in the figure captions.

The pitch length for the springs needs to reported with the errors.

In 'fitting model of the force-extension curves' section, the Hooke's law described as $F= Kx+x0$ is missing the parenthesis and it should be written as $F= K(x+x0)$.

AFM Images in S4a looks out of focus and blurry. Please ignore this comment if this is an artifact of document compression. Otherwise, please address the issue.

Reviewer #3 (Remarks to the Author):

The work presents carefully designed experiments and molecular dynamics simulations to investigate the effect of chemomechanical forces on "DNA origami nanosprings". The results of the experiments and the simulations are interesting and in agreement.

The combination of experimental work using optical tweezers and AFM together with the molecular dynamics is interesting to conclusively show that chirality is determined by rotational torques in DNA nanosprings. The wider implication of this finding for similar-scale biomolecules is significant for the field.

The manuscript is clearly written, and discusses the results in good detail. I enjoyed reading the manuscript and would be happy to recommend for its publication after the authors could address the following:

1. The investigated structure is presented, but the text assumes the readers should know what "DNA origami nanosprings" are. A more general description of the structure and the concept is required in the introduction especially for a generalist journal.

2. The opening of the abstract and the introduction mentions replisomes and ribosomes, but the work is specifically focused on DNA structures. The proposition that DNA structures to be models for other mesoscale biomolecular assemblies is vague. I suggest a more direct introduction and clarification of the aim of the work.

3. The second sentence of the abstract is too strong, neglecting a literature of work. Third sentence of the introduction is a similar sentence with a more balanced tone. Consistency is needed.

4. Third paragraph of the introduction discusses long-range molecular interactions and arrangements. The text would benefit a discussion on allostery as a long-range mechanism.

5. Is there a particular reason for the selection of a DNA structure with 37 modules? Would the results be scalable/similar for structures with different numbers of modules?

6. The reason for selection of different linkers, i.e. biotin/streptavidin and digoxigenin/anti-digoxigenin should be explained.

7. Figure 3d-e suggest two different regimes with a threshold force level of 2 pN. However, the force measurement resolution is not clear. How accurate (in terms of absolute and incremental resolution) can the force be measured/controlled? What is the effect of force uncertainty?

8. Discussion about the right-handedness vs. left-handedness based on AFM imaging is clear. However, the assessment of slopes (Figure 4) should be directly influenced by the tip geometry and the scan parameters (mode, scan direction, etc.). These should be added in the manuscript. Are these measurements repeatable if scan direction is changed?

Reviewer #4 (Remarks to the Author):

In this manuscript, the authors use optical tweezers to study the mechanical properties of a mesoscale spring created using DNA origami techniques. This particular “nanospring” has the unique property of being able to switch between different coiled states in response to the presence of different “molecular actuators”, i.e. K⁺ or DNA oligos with specific sequences. They observe that while switching between these can flip the orientations of local DNA junctions, it does not change the right-handed chirality of

the overall nanospring structure. The data is supported with MD simulations and AFM imaging. The authors then try to generalize the observations to determine overall principles of mesoscale structure.

While the study is interesting, and the experiments and analysis are carried out competently, it is not clear whether these results have any broad applicability, or if simply apply to this very specific model. Correspondingly, it is clear whether these results, which are somewhat technical and specific, would be of general enough interest for the readership of Nature Communications. For example, in the abstract it is stated that the results “provides evidence that mesoscale helical handedness is governed by the torque, rather than the achiral orientation, of nanospring backbones”, but the manuscript does not provide a convincing case that this is true in any general or broadly applicable way, outside of the carefully engineered DNA origami system under study. It would help if the authors could demonstrate that these principles also hold in other mesoscale systems, or at least perform some analysis that directly and specifically connect their results to other mesoscale systems that would likely behave in a similar way.

Regarding the nanospring studied in this manuscript, it is an interesting construct that can be actuated in different ways into different states. However, the novelty of the construct and the experiments, and therefore of the manuscript, is lessened by multiple previous publications from this group that describe both very similar molecular constructs and their combination with optical tweezers force experiments [references 16 and 17].

Regarding the force spectroscopy results, the analysis would benefit from an attempt to bridge the force-extension measurements with the force jump experiments. For example, while the force-jump experiments were used to determine the effective spring constant of the nanospring, this information should also be encoded in the slope of the force extension curves. It would be helpful if a comparison between these measurements were presented. For example, in the force jump experiments, the spring constant is only discussed for the under 2 pN regime and the over 2 pN regime, but from the force extension data it looks like this should be more of a continuum. It would be helpful if this were discussed quantitatively. Understandably there may be some complications due to hysteresis and the dynamics of the system, but these could be discussed and analyzed as well.

Increased discussion and analysis would also benefit some of the other claims in the manuscript, such as how the handedness is determined primarily by “the torque, rather than the achiral orientation”. It would be helpful if this was better explained, ideally with a connection to the mechanics of the pulling experiments.

Overall, this is an interesting article that would benefit from additional discussion and analysis, though based on impact and general interest arguably suitable for a more field-specific journal than Nature Communications.

REVIEWER COMMENTS

Reviewers #1 and #2, who have reviewed jointly (Remarks to the Author)

The authors have described a DNA origami nanospring with two bending mechanisms that can alter the properties of the spring. This improved version of their earlier published nanosprings exhibits a responsive structure and the authors characterize its spring constant using force jump experiments. Interestingly, the nanospring maintains its unique handedness even when its backbone is flipped inside-out which the authors attribute to the inherent rotational torque in DNA. The potential for numerous nanomechanical applications makes this nanospring an exciting discovery. However, there are a few concerns that the authors must address before publication:

> The writing of the manuscript included a significant number of grammatical errors and this contributed to making the readability of the manuscript fairly poor. Moreover, the text was dense and at times difficult to follow. So I would highly encourage authors to proof-read the text and attempt to make the writing more concise.

→ We have revised the manuscript with necessary changes.

> The histogram in figure 1d has a poor fit (indicated in red) that dips below zero near 10 and 40 nm. The data binning in the histogram seems too wide and could be the cause of the poor fit. The authors are encouraged to narrow the bin width and, if necessary, add additional data points from AFM measurements to improve the quality of the fit.

→ We have revised histograms (Figure 1d, e) and replaced Figure 1f with new Figures 1f and g for the number of turns in different nanosprings (see next point below).

Figure 1. Design of a dual-switching nanospring. (a) Reversible transformation of a DNA origami bundle into a spring shape through the cumulative actuation of K^+ -responsive modules. The details of the module (dashed red box) are shown in (b) and Supplementary Figure S2. (b) Schematics of the module. The ssDNA bridge strand containing a G-rich sequence (5'-GGGTTAGGGTTAGGGTTAGGG-3') flanked with staple sequences is incorporated in the module. The strand forms a G-quadruplex in the presence of K^+ , which leads to the bending of the module. The strand can also hybridize with an anti-GQ strand carrying a toehold sequence (5'-CCCTAACCTAACCTAACCCAGAGAACT-3'). The 21 bp duplex induces the bending whose direction is opposite to that induced by the GQ-formation. The anti-GQ strand can be displaced from the module via the toehold-mediated strand displacement process with a releaser strand, 5'-

AGTTCTCTGGGTTAGGGTTAGGGTTAGGG-3', whose sequence is fully complementary to the anti-GQ strand. (c) Representative AFM images of the nanospring taken after the hybridization with the anti-GQ strand (left), in the absence of both the anti-GQ strand and 100 mM KCl (middle), and the presence of 100 mM KCl (right). All imaging was performed in 5 mM pH 8.0 Tris buffer with 15 mM MgCl₂ and 1 mM EDTA, with or without KCl at room temperature. (d, e) Histograms of the curvature radius of the nanospring after hybridization with anti-GQ strands (d) and that in the presence of 100 mM KCl without bound anti-GQ (e). *N* represents the total number of nanospring molecules evaluated. (f, g) Number of turns calculated from the curvature radius and contour length for nanospring measured by AFM after hybridization with anti-GQ strands (f) and that in the presence of 100 mM KCl without bound anti-GQ (g). “+/-” signs represent standard deviations.

> The nanospring turn calculation seems inconsistent in the GQ-NS and anti-GQ-NS nanosprings (estimated below). Further, the authors use a different turn number for the anti-GQ-NS spring in page 8, where it is claimed to be 3.2 turns. The authors must rectify this discrepancy and also report the exact contour length (with error if possible) used in the calculations. The contour length must be supported with citations or measured from existing AFM images.

GQ-NS nanospring: $550\text{nm}/(2 \times 3.14 \times 45.0\text{nm}) = 1.94$ turns (reported as 1.8 ± 0.4).

anti-GQ-NS spring: $550\text{nm}/(2 \times 3.14 \times 26.7\text{nm}) = 3.3$ turns (reported as 3.4 ± 0.4).

→ We apologize for our insufficient information. We have calculated the number of turns from the estimated contour length (see histograms below, now added as Supplementary Figure S6) for individual nanosprings and then obtained mean +/- SD (Revised Figure 1f, g). We have revised Figure 1 and described values in the figure caption. To measure the contour lengths, we approximated the structure by taking 30 points along its apparent shape. A notable point is that there is a significant difference between the value of anti-GQ-NS and that of GQ-NS. We believe this difference is possibly due to the opposite mechanism for the transformation, i.e., pulling by GQ formation and the pushing effect of duplex formation. Considering the above point, calculating number of turns using theoretical length of 550 nm may not be accurate as caDNAo assumes 1 bp = 0.34 nm and does not consider any possible changes by bending and twisting. Based on this, we obtain the average turns for GQ-NS nanospring as $487\text{ nm}/(2 \times 3.14 \times 43.9\text{ nm}) = 1.8$ turns and those for anti-GQ-NS nanospring as $554\text{ nm}/(2 \times 3.14 \times 25.8\text{ nm}) = 3.4$ turns, which are almost identical to those (1.7 and 3.5 nm respectively, (page 7 in the main text)) measured from individual nanosprings (Figure 1f and 1g).

Figure S6: Histograms representing the contour length of (a) antiGQ-NS and (b) GQ-NS. The data have been collected by estimation from the apparent shape in AFM images using 30 points along the shape (see Figure S5).

> In the force jump experiment the authors describe that “For GQ-NS, the 10mM Tris-buffer condition was maintained with 100 mM KCl while for anti-GQ-NS, it was maintained with 100 mM LiCl.” This seems to contradict the single-molecule force ramping assay where anti-GQ-NS was stretched in a solution of 1 μ M anti-GQ oligos in 100 mM LiCl. If the authors did perform the experiment without an excess of anti-GQ oligos, the authors need to demonstrate that the anti-GQ oligos on the nanosprings do not denature under force. If possible, the authors should further quantify the number of cycles that the nanospring can be stretched and relaxed without compromising its structural properties.

→ For the force-jump experiments, we used purified anti-GQ nanospring samples such that no surplus oligos should interfere with the structures formed in the junctions of nanosprings. However, to rule out any discrepancies as suggested by the reviewers 1+2, we did force jump assays with the purified anti-GQ-NS as well as with 1 μ M anti-GQ oligo, which is presented below. No significant difference on the kinetics and spring constants was observed.

Figure R1: Recoil and uncoil kinetics for the antiGQ-nanospring without and with 1 μM anti-GQ oligo. Both conditions showed similar recoiling and uncoiling kinetics, as well as spring constants measured from recoiling and uncoiling events.

→ Previous literatures (Phys. Rev. E 78, 011920 (2008), Phys. Rev. E 84, 031905 (2011), Proc. Natl. Acad. Sci. 96, 20 (1999), 11277–11282) have mentioned that the shearing force to disrupt a duplex DNA lies in range 30-55 pN for 12-20 bp length. With a high GC content (57%) in our GQ-antiGQ duplex, we believe the longitudinal denaturing force is above 30 pN. Also, we previously (Nucleic Acids Res. 44, 14 (2016) 6574–6582) found that for origami nanoassemblies, the longitudinal force for the disassembly of staples and Holliday junctions is ~35 pN. Since the experiments were performed below 30 pN in current work, we expect the duplexes are not denatured. This can also be confirmed from reversible FX curves obtained in force-ramping experiment copied below.

Figure S9: Multiple overlapped force-extension stretching curves for the same antiGQ-nanospring. Subsequent traces following the same path during stretching of the nanospring up to 30 pN force indicate no structural deformations of the nanospring occurred during repetitive force-ramping processes.

→ As per reviewers' suggestion, we also carried out multiple rounds of force-jump assays to check the stability of nanosprings. At least 3 sets of force-jump experiments were successfully carried out from the same GQ-nanospring molecule without any structural compromise. A typical plot is shown below.

Figure S10: Three consecutive sets of force jump experiments carried out for the same GQ-nanospring. Each set consists of the force jump from 30 to 0.5, 1, 2...10 pN as well as from 0.5 pN to 1, 2, 3...10 pN.

→ We added a paragraph to discuss these points on page 12. "It is noteworthy that nanosprings can maintain their structural integrity in the force range we have applied (up to 30 pN). First, consecutive force-extension curves revealed overlapping stretching and relaxing traces in the same nanospring construct (Supplementary Figure S9). Second, consecutive force jump experiments also displayed similar uncoiling and recoiling events for the same nanospring (Supplementary Figure S10). Both experiments indicated intact nanospring structures under 30 pN force."

> Could the recoiling and uncoiling rate of nanosprings change with KCl or oligo concentration? The authors need to elaborate further on this dependence. How did the authors arrive at the concentrations reported in the experiments?

→ We intended to mimic the cellular K⁺ concentration of 100 mM (Pediatric Nephrology, Springer, 2009, 185–204) for our study. Previous research (Biophysical Chemistry, 211, (2016), 70-75) has revealed that changes in KCl concentration indeed causes conformational changes in G-quadruplexes. As suggested by the reviewer, we varied the KCl concentration to 500 mM and repeated force-jump assay. We have found different kinetics compared to 100 mM KCl, which is presented below.

Figure R2: Recoiling and uncoiling kinetics for the GQ-nanospring with 500 mM KCl concentration.

→ We believe the discrepancy in kinetics under different KCl concentrations may be due to the mesoscopic mechanical properties of DNA bundles since an increased salt concentration resulted in more rigid DNA backbones. In addition, under different salt concentrations, the conformation of the G-Quadruplex formed in the junction between neighboring piers may alter, which not only results in different end-to-end distance of the bridge strands, but also affects the coiling/uncoiling processes, thereby exhibiting different kinetics. To test this, we performed mechanical unfolding experiments with the construct that contains only one G-Quadruplex (5'-TTAGGGTTAGGGTTAGGGTTAGGGTTA-3', which bears the same core GQ forming sequence used in current work) under 100 mM or 500 mM KCl in a 10 mM Tris buffer (pH 7.4) at room temperature. We found a single unfolding force population of ~22 pN at 100 mM KCl. In contrast, we found multiple unfolding force populations of ~15 pN, 26 pN, 40 pN, and 54 pN, suggesting the existence of several conformations of G-quadruplexes at 500 mM KCl. Given that the 100 mM KCl buffer condition would minimize the complexity in nanospring structure, we used this condition in current work.

Figure R3: Unfolding force histograms of the telomeric G-Quadruplex under (a) 100 mM KCl and (b) 500 mM KCl concentrations using a setup similar to the literature (Biochemistry, 59, 37 (2020), 3438–3446). N represents the number of traces and n represents the number of molecules.

→ For the concentration of antiGQ oligos, we rationalized from AGE gel images (see Figure S3) where a ratio of 1:1000 (GQ-NS: anti-GQ oligos) produced a much more distinct anti-GQ-NS compared to GQ-NS. Thus, we chose 1 μ M anti-GQ oligo concentration, since the concentration of origami used in optical tweezers set-up is in nanomolar range. (Nucleic Acids Res. 41, 6 (2013), 3915–3923).

→ We have addressed the rationale in page 5, which is copied below, “To ensure complete hybridization, we used 1 μ M anti-GQ strand, which is about 1000 times higher concentration (see Supplementary Figure S3 for optimized hybridization ratio) than the effective concentration of single molecules tethered between two trapped particles.²⁴”

> It is unclear if the force jumps performed in figure 3a were done on GQ-NS or anti-GQ-NS. The authors have reported only one of the nanospring temporal traces for the force jumps. The force jump data for the second nanospring could be added to the supplementary information.

→ We thank the reviewers for pointing it out. Figure 3a was done on GQ-NS. We have revised the caption in Figure 3a which represents a typical force-jump plot for the GQ-NS and included a typical force-jump plot of antiGQ-NS in SI which is copied below.

Figure S8: Temporal traces of forces and extensions during different force-jump events in antiGQ nanospring.

> In figure S3, lane 4 seems to have a higher molecular weight structure than the initial nanospring as well as other lanes when added with 1:100 fold excess of releaser strand. Is this an artifact or reproducible phenomenon? The authors need to describe the lane components in further detail than it has already been done in the figure captions.

→ This is a reproducible phenomenon. We interpret that the observed band shift reflects the increase in the molecular weight due to the formation of an intermediate complex of the strand displacement where both anti-GQ and releaser strands remain bound to an GQ-forming strand. We have mentioned this point in the revised figure captions. Experimental flow and the lane components were also described in more detail in the caption. The revised Figure S3 and its caption is copied below.

- Samples:
M. 1 kbp ladder
S. p8064
1. NS-GQ21
2. NS-GQ21 + anti-GQ21-SD8 (NS-GQ21 : anti-GQ21-SD8 = 1:1000 [after purification])
3. [NS-GQ21 + anti-GQ21-SD8] : GQ21-SD8 = 1 : 100
4. [NS-GQ21 + anti-GQ21-SD8] : GQ21-SD8 = 1 : 1000
5. [NS-GQ21 + anti-GQ21-SD8] : GQ21-SD8 = 1 : 10000

Figure S3: Agarose gel electrophoresis (AGE) and AFM analyses of reversible switching of the nanospring shape. (a) AGE analysis. To make the anti-GQ nanospring (anti-GQ-NS), anti-GQ strands (ASs) were first added at the molar ratio of nanospring : anti-GQ strand = 1 : 1000. The excess volume of unincorporated anti-GQ strands was then removed by the PEG-sedimentation-based purification. After that, releaser strands (RSs) were added at the molar ratio of nanospring : releaser strand = 1 : 100, 1 : 1000, or 1 : 10000. M: 1 kbp ladder marker; S: p8064 scaffold; lane 1: NS before incubation with AS; lane 2: NS after incubation with anti-GQ strand AS (anti-GQ-NS); lanes 3–5: anti-GQ-NS after incubation with different concentrations of releaser strand. Anti-GQ-NS: releaser = 1:100 (lane 3), 1:1000 (lane 4), and 1:10000 (lane 5). Excess amount of RSs was observed as the lower bands in lanes 3-5. The higher band observed in lane 4 may reflect the formation of an intermediate complex in which both anti-GQ and releaser strands remain bound to a GQ-forming strand. (b) AFM images of sample loaded into Lane 3 of Figure S3 (a). (c) Reversible switching of the nanospring shape via toehold-mediated strand displacement reaction. Representative AFM images of nanospring before (left) and after incubation with AS (middle) and after subsequent incubation with releaser strand (RS) (right) are shown. (d) Reversible switching of the nanospring shape via folding/unfolding of G-quadruplex. Representative AFM images of nanospring before (left) and after incubation with 100 mM KCl (middle) and after subsequent removal of KCl from the sample (right) are shown. The reversible transformation of the construct with GQ-forming sequences was achieved by changing the KCl concentration in the buffer from 0 and 100 mM. A 100 kDa MWCO centrifuge filter (Amicon Ultra, Merck Millipore, Billerica) was used to exchange the buffer.

The pitch length for the springs needs to be reported with the errors.

→ We have revised the pitch lengths with updated calculations and standard deviations in Page 9 which are copied below.

“Since the numbers of coils in the two nanosprings are 1.7 ± 0.4 and 3.5 ± 0.4 for GQ-NS and anti-GQ-NS, respectively (Figure 1f, g), pitch lengths for these nanosprings were estimated as 189 ± 9 nm/ 1.7 turns = 110 ± 30 nm and 138 ± 3 nm/ 3.5 turns = 39 ± 5 nm in 3D space.”

> In ‘fitting model of the force-extension curves’ section, the Hooke's law described as $F = Kx + x_0$ is missing the parenthesis and it should be written as $F = K(x + x_0)$.

→ We apologize for the error, which has now been corrected in SI Page 7.

> AFM Images in S4a looks out of focus and blurry. Please ignore this comment if this is an artifact of document compression. Otherwise, please address the issue.

→ Figure S4 has been replaced with a clear version which along with caption is copied below.

Figure S4: Representative AFM images of nanosprings. (a) Anti-strand-incorporated nanospring (anti-GQ-NS). (b) G-quadruplex-induced nanosprings (GQ-NS). (c) Nanospring (GQ-NS) in the presence of 100 mM of LiCl.

Reviewer #3 (Remarks to the Author):

> The work presents carefully designed experiments and molecular dynamics simulations to investigate the effect of chemomechanical forces on “DNA origami nanosprings”. The results of the experiments and the simulations are interesting and in agreement. The combination of experimental work using optical tweezers and AFM together with the molecular dynamics is interesting to conclusively show that chirality is determined by rotational torques in DNA nanosprings. The wider implication of this finding for similar-scale biomolecules is significant for the field. The manuscript is clearly written and discusses the results in good detail. I enjoyed reading the manuscript and would be happy to recommend for its publication after the authors could address the following:

1. The investigated structure is presented, but the text assumes the readers should know what “DNA origami nanosprings” are. A more general description of the structure and the concept is required in the introduction especially for a generalist journal.

→ We thank the reviewer 3 for the suggestions. We have included some text in Page 3 in the introduction which is copied below.

“A typical DNA origami nanoassembly employs conventional Watson-Crick base pairing in which several DNA duplexes are bundled together to form a 2D or 3D nanostructures.⁴⁻⁷ The method is assisted by computer aided designs to simulate hybridization of several tens to hundreds of single-stranded small DNA fragments, called staples, onto a long single-stranded scaffold template DNA.^{3,8} Such a one-pot annealing reaction readily synthesizes higher order nano and mesoscale structures. With the precise and specific base pairing in DNA duplexes and supramolecular nature of DNA origami self-assembly, the method provides ample space to introduce different functional groups.”

2. The opening of the abstract and the introduction mentions replisomes and ribosomes, but the work is specifically focused on DNA structures. The proposition that DNA structures to be models for other mesoscale biomolecular assemblies is vague. I suggest a more direct introduction and clarification of the aim of the work.

→ We want to clarify that mentioning of replisomes and ribosomes in text represented examples of mesoscale structures that are present inside cells and the work carried out here is to use DNA based origami to mimic mesoscale structures, which may or may not be a replisome or ribosome. To gain clarity, we are no longer specifically mentioning the ribosome or replisome anymore. The revised texts are copied below.

In abstract: “Many cellular machineries in the biological framework are mesoscale (10-100 nm) molecular assemblies.”

In introduction: “In the biological context, the mesoscale dimensions hold an important locus as most viral particles and many important cellular machineries are mesoscale sized biomolecular assemblies.”

> 3. The second sentence of the abstract is too strong, neglecting a literature of work. Third sentence of the introduction is a similar sentence with a more balanced tone. Consistency is needed.

→ We have modified the sentence in abstract as follows.

“Due to the limited characterization tools, intrinsic complexity of heterogenous subcellular components, paucity of available data, and inadequate model systems, underlining principles for topological arrangement and transition dynamics of these mesoscale structures have rarely been elucidated.”

> 4. Third paragraph of the introduction discusses long-range molecular interactions and arrangements. The text would benefit a discussion on allostery as a long-range mechanism.

→ This is a great point. We have included a sentence about allostery in Page 4 of the manuscript which is copied below.

“Intermolecular force (IMF)¹⁸ can induce conformational variation in different parts of a protein, leading to allostery in a nanometer scale.^{19,20} However, in mesoscale helices, IMF may not be strong enough to sustain the preferential long-range molecular arrangement across hundreds of nanometers space to produce different helical senses (i.e., left-handed or right-handed twists) in the mesoscopic chirality.”

> 5. Is there a particular reason for the selection of a DNA structure with 37 modules? Would the results be scalable/similar for structures with different numbers of modules?

→ The nanospring design has been conceptualized using caDNA based on a long DNA scaffold template of p8064. The size of p8064 limits the number of modules to 37. To the best of our knowledge, p8064 is the longest commercially available scaffold DNA. However, when a longer scaffold DNA is used, we can readily be able to scale into larger (preferably longer than our current length of ~550 nm) nanospring sizes.

> 6. The reason for selection of different linkers, i.e. biotin/streptavidin and digoxigenin/anti-digoxigenin should be explained.

→ We have included the following text in Page 4 of Supplementary Information under the heading, “Preparation of the poly-Digoxigenin and Biotin labeled dsDNA handles”.

“Biotin/streptavidin and digoxigenin/anti-digoxigenin linkages have been extensively used for decades in force-based single molecule assays because of their high specificity, binding affinity and force stability.⁶ Although single digoxigenin/anti-digoxigenin interaction has lower force stability (~25 pN) compared to biotin/streptavidin interaction (~200 pN), multiple digoxigenin to anti-digoxigenin interactions significantly increase the mechanical stability. Moreover, use of different linkers at the end of DNA handles increases the tethering yield of individual nanospring constructs to different polystyrene beads for single-molecule assays.”

> 7. Figure 3d-e suggest two different regimes with a threshold force level of 2 pN. However, the force measurement resolution is not clear. How accurate (in terms of absolute and incremental resolution) can the force be measured/controlled? What is the effect of force uncertainty?

→ The force resolution in our optical tweezers was found to be 0.1 pN (*J. Am. Chem. Soc.*, 136, (2014), 40, 13967–13970).

> 8. Discussion about the right-handedness vs. left-handedness based on AFM imaging is clear. However, the assessment of slopes (Figure 4) should be directly influenced by the tip geometry and the scan parameters (mode, scan direction, etc.). These should be added in the manuscript. Are these measurements repeatable if scan direction is changed?

→ As per the reviewer’s suggestion, we conducted AFM scanning through two directions at different positions. Information regarding the tip geometry and scan mode has been added in “Materials and methods” section under the heading “AFM observation”, which is copied below.

“Small cantilevers (9 μm long, 2 μm wide, and 100 nm thick) with an electron-beam-deposited carbon tip (tip length: ~ 2 μm, tip radius: < 10 nm) having a spring constant of 0.1 N/m and a resonant frequency of

~300–600 kHz in water (USC-F0.8-k0.1-T12; Nanoworld, Neuchâtel, Switzerland) were used to scan the sample surface. The 320 × 240-pixel images were collected at a scan rate of 0.5 frames per second (fps) with tapping mode.”

→ Both scanning directions provided us with the same right-handedness of nanosprings. We have included the data as a new Supplementary Information Figure S13, which is copied below.

Figure S13: Different scanning directions of AFM for a) antiGQ-NS and b) GQ-NS. Both scanning directions (either from left to right or from right to left) revealed right-handed chirality of nanosprings.

→ We have added a sentence in Page 13 to clarify this point, “When we varied the scan directions of the AFM tip, we also observed the same chirality in both nanosprings (Supplementary Figure S13).”

> Reviewer #4 (Remarks to the Author):

In this manuscript, the authors use optical tweezers to study the mechanical properties of a mesoscale spring created using DNA origami techniques. This particular “nanospring” has the unique property of being able to switch between different coiled states in response to the presence of different “molecular actuators”, i.e. K⁺ or DNA oligos with specific sequences. They observe that while switching between these can flip the orientations of local DNA junctions, it does not change the right-handed chirality of the overall nanospring structure. The data is supported with MD simulations and AFM imaging. The authors then try to generalize the observations to determine overall principles of mesoscale structure.

> While the study is interesting, and the experiments and analysis are carried out competently, it is not clear whether these results have any broad applicability, or if simply apply to this very specific model. Correspondingly, it is clear whether these results, which are somewhat technical and specific, would be of general enough interest for the readership of Nature Communications. For example, in the abstract it is stated that the results “provides evidence that mesoscale helical handedness is governed by the torque, rather than the achiral orientation, of nanospring backbones”, but the manuscript does not provide a convincing case that this is true in any general or broadly applicable way, outside of the carefully engineered DNA origami system under study. It would help if the authors could demonstrate that these principles also hold in other mesoscale systems, or at least perform some analysis that directly and specifically connect their results to other mesoscale systems that would likely behave in a similar way.

→ Considering the broad applications of DNA origami-based nanostructures, which have been demonstrated in work published in *Nat Commun* **7**, 13715 (2016), *Commun Biol* **2**, 437 (2019), *Nat Commun* **13**, 3182 (2022), etc. on monitoring the surface proteins like actins, myosins and microtubules, many of which are chiral mesoscale assemblies, we believe our findings will provide a fundamental understanding of structural elements involved in mesoscale chirality, as well as more practical applications of chirality interactions between mesoscale structures (*Nat Commun* **13**, 76 (2022)).

→ In addition, our nanospring is constructed as a 6-helix bundle (6HB), one of the simplest and thus fundamental 3D DNA origami structure packed on honeycomb-lattice geometry. Given that a variety of 3D DNA origami nanomachines have been constructed based on honeycomb-lattice geometry (*Science* **335**, 837 (2012), *ACS Nano* **13**, 5, 5959–5967 (2019), *Nat Commun* **13**, 3182 (2022), etc.), our findings possess general impact and broad applicability in designing future DNA nanomachines that exhibit more complex motions involving bending and twisting of their shapes.

→ As suggested by reviewers, we tone down the statement of “provides evidence that mesoscale helical handedness is governed by the torque, rather than the achiral orientation, of nanospring backbones” to “provides evidence that mesoscale helical handedness **may be** governed by the torque...”. In addition, we have synthesized a left-handed nanospring origami based on a skipped version in nanospring backbones, this provides a strong support for the hypothesis that mesoscale helical handedness is governed by torque (This set of data is beyond the scope of current work; it will be reported in due course, hopefully in *Nat. Commun.* as well).

> Regarding the nanospring studied in this manuscript, it is an interesting construct that can be actuated in different ways into different states. However, the novelty of the construct and the experiments, and therefore of the manuscript, is lessened by multiple previous publications from this group that describe both very similar molecular constructs and their combination with optical tweezers force experiments [references 16 and 17].

→ We agree that similar DNA origami was self-assembled previously by our labs, but the overall theme of this manuscript is entirely different from the previous work. In this paper, we have emphasized that inherent helicity of the duplex DNA is the major player which determines the mesoscale chirality. The novelty of this manuscript has also been recognized by other reviewers as we found that chirality is not influenced by the inside-out flipping of the nanospring backbone. We wish to stress that this behavior is counter-intuitive and never reported before. We would also like to mention that we have successfully prepared a new DNA nanospring construct that varied the helical sense to the opposite left-handed helicity based on the torque manipulation of the duplex DNA bundle (see the point above). In addition, using 2D AFM imaging to determine 3D helical chirality (Figure 4) is innovative. It provides a generic approach to evaluate 3D chirality without resorting to highly involved techniques such as X-ray or NMR.

> Regarding the force spectroscopy results, the analysis would benefit from an attempt to bridge the force-extension measurements with the force jump experiments. For example, while the force-jump experiments were used to determine the effective spring constant of the nanospring, this information should also be encoded in the slope of the force extension curves. It would be helpful if a comparison between these measurements were presented. For example, in the force jump experiments, the spring constant is only discussed for the under 2 pN regime and the over 2 pN regime, but from the force extension data it looks like this should be more of a continuum. It would be helpful if this were discussed quantitatively. Understandably there may be some complications due to hysteresis and the dynamics of the system, but these could be discussed and analyzed as well.

→ We appreciate reviewer 4 raised this constructive suggestion. We agree that force-extension curve may also provide the mechanical information of nanosprings. We have obtained spring constants by fitting force-extension curves using an equation combining the worm-like-chain model and Hooke's law (Figure S7). We found that the spring constant values (0.04 and 0.03 pN/nm for GQ-NS and anti-GQ-NS, respectively, fit within the force range of 0.1-8.2 pN) are located between those obtained from the recoiling force jump experiments < 2 pN (0.02 and 0.03 pN/nm for GQ-NS and anti-GQ-NS, respectively) and those at 2-10 pN (0.41 and 0.56 pN/nm for GQ-NS and anti-GQ-NS, respectively). It is noteworthy that recoiling jump experiments can be monitored with longer time than uncoiling force jumps, therefore, it is more reliable to retrieve the spring constants during recoiling force jump experiments. The consistent results obtained by force jump and force extension experiments validate the spring constant measurements. However, we believe the force jump experiments are more accurate based on the following reasons.

1) In our construct, we sandwiched the nanospring between two long (each 2520 bps equaling to a total of ~2.3 μm) DNA handles (see Figure 2a). In slow and continuous force extension curves, the effective mechanical quantifications of the whole construct (nanospring + DNA handles) are mainly contributed from those long duplex DNA handles, instead of the GQ or the GQ-antiGQ duplex (<10 nm) formed in the junction of the neighboring piers in the nanospring. In contrast, force jump assay provides a unique

perspective to effectively differentiate the mechanical quantifications (such as spring constants) between the dsDNA handles and the DNA nanospring. These mechanical properties are respectively determined by the transition kinetics in the DNA handles and GQ (or GQ-antiGQ duplex) in DNA nanospring junctions in response to force variations (see Figure 3b&c for their much different transition kinetics).

2) The force-extension curve fitting assumes that the spring constant of the nanospring does not vary in the force range we fit. However, this may not be true. The spring constant of the nanospring may not be constant over a force range since DNA nanosprings are soft and their conformation is expected to vary with force. The force jump experiment can precisely evaluate the recoiling and uncoiling events at a particular force. Therefore, it is more accurate to profile the spring constant against different force ranges. In fact, this is what we observed in Figure 3, where two distinct regions were found for two different spring constants.

→ Now, we added this discussion in page 11-12, which is copied below.

“As observed from Figure 3b&c, the recoiling events (time to reach recoiling equilibrium) are much longer than uncoiling events, hence, it is more reliable to retrieve the spring constants in the recoiling, instead of uncoiling force jumps. We also measured spring constants by fitting force-extension curves with an equation²² combining the worm-like chain model with the Hooke’s law (see SI). We found that the spring constant values (0.04 and 0.03 pN/nm for GQ-NS and anti-GQ-NS, respectively, when fitted within the force range 0.1-8.2 pN) are located between those obtained from the recoiling force-jump experiments at < 2 pN (0.02 and 0.03 pN/nm for the GQ-NS and anti-GQ-NS, respectively) and those at 2-10 pN (0.41 and 0.56 pN/nm for the GQ-NS and anti-GQ-NS, respectively). Both methods therefore validated the spring constant measurements. However, the force-jump experiment is more accurate as it can precisely evaluate the recoiling and uncoiling events at a particular force. In contrast, force-extension fitting assumes that the spring constant does not vary in the fitted force range, which may not be true as conformation of soft DNA nanosprings is expected to change with force. In addition, in the slow and continuous force extension curves, the effective mechanical quantifications of the whole construct (nanospring + DNA handles, see Figure 2a)) are mainly contributed from long duplex DNA handles (~ 2.3 μm), instead of the GQ or the GQ-antiGQ duplex (<10 nm) formed in each junction of the neighboring piers in the nanospring. On the other hand, force jump assay provides a unique perspective to effectively differentiate the mechanical quantifications (such as spring constants) between the dsDNA handles and the DNA nanospring, which are respectively based on the transition kinetics in the DNA handles and GQ or GQ-antiGQ duplex in nanospring junctions in their responses to rapid force variations (see Figure 3b&c).”

> Increased discussion and analysis would also benefit some of the other claims in the manuscript, such as how the handedness is determined primarily by “the torque, rather than the achiral orientation”. It would be helpful if this was better explained, ideally with a connection to the mechanics of the pulling experiments.

→ We hypothesize that the handedness results from the inherent torque of the duplex DNA, however, pulling experiments cannot give accurate predictions as force is not chiral. For that, we have successfully changed the helicity of the nanospring (left-handed) by incorporating a skip version of duplex DNA bundles in the origami (see under the first point of Reviewer 4). In this manuscript, we tone down the sentence as

“This result provides evidence that mesoscale helical handedness **may be** governed by the torque, rather than the achiral orientation, of nanospring backbones.”

> Overall, this is an interesting article that would benefit from additional discussion and analysis, though based on impact and general interest arguably suitable for a more field-specific journal than Nature Communications.

→ We believe this manuscript comprises two intensively investigated research areas; 1. DNA origami system and 2. Chirality. DNA origami system has flourished over the past decades due to their remarkable applicability and tunability, here, we 1) presented it as a model system for mesoscale structures, and 2) demonstrated that the overall chirality of the nanospring is not based on the orientation of the DNA backbone. We should stress that currently, there is a lack of easy-to-prepare and versatile model system with nanometer assembling precision to investigate the fundamental aspects of mesoscale structures that are important biological machineries. In addition, by addressing all the points from reviewers, we have added additional results, analyses, and discussions to strengthen the innovation and the significance of this work. We believe the findings presented here would gather the interest of a large audience.

REVIEWERS' COMMENTS

Reviewer #1 (Remarks to the Author):

The authors have now addressed my previous comments and I am satisfied.

Reviewer #2 (Remarks to the Author):

The authors have sufficiently answered the concerns and have revised the manuscript in response to the issues described in the past. It is now suitable for publishing.

Reviewer #3 (Remarks to the Author):

The authors have satisfactorily addressed all the points that I and other reviewers raised including new experiments. The revised manuscript is significantly better, and I am happy to recommend its publication.

REVIEWER COMMENTS

Reviewer #1 (Remarks to the Author):

The authors have now addressed my previous comments and I am satisfied.

→ *Thank you for all the insightful comments.*

Reviewer #2 (Remarks to the Author):

The authors have sufficiently answered the concerns and have revised the manuscript in response to the issues described in the past. It is now suitable for publishing.

→ *Thank you for all the insightful comments.*

Reviewer #3 (Remarks to the Author):

The authors have satisfactorily addressed all the points that I and other reviewers raised, including new experiments. The revised manuscript is significantly better, and I am happy to recommend its publication.

→ *Thank you for all the thoughtful comments.*